# Modeling apical and basal tree contribution to orientation selectivity in a mouse primary visual cortex layer 2/3 pyramidal cell

Konstantinos-Evangelos Petousakis[1,2†], Jiyoung Park[3†], Athanasia Papoutsi[2], Stelios Smirnakis[3]*, Panayiota Poirazi[2]*

[1]Department of Biology, University of Crete, Heraklion, Greece; [2]IMBB, FORTH, Heraklion, Greece; [3]Department of Neurology, Brigham and Women's Hospital and Jamaica Plain Veterans Administration Hospital, Harvard Medical School, Boston, United States

**Abstract** Pyramidal neurons, a mainstay of cortical regions, receive a plethora of inputs from various areas onto their morphologically distinct apical and basal trees. Both trees differentially contribute to the somatic response, defining distinct anatomical and possibly functional sub-units. To elucidate the contribution of each tree to the encoding of visual stimuli at the somatic level, we modeled the response pattern of a mouse L2/3 V1 pyramidal neuron to orientation tuned synaptic input. Towards this goal, we used a morphologically detailed computational model of a single cell that replicates electrophysiological and two-photon imaging data. Our simulations predict a synergistic effect of apical and basal trees on somatic action potential generation: basal tree activity, in the form of either depolarization or dendritic spiking, is necessary for producing somatic activity, despite the fact that most somatic spikes are heavily driven by apical dendritic spikes. This model provides evidence for synergistic computations taking place in the basal and apical trees of the L2/3 V1 neuron along with mechanistic explanations for tree-specific contributions and emphasizes the potential role of predictive and attentional feedback input in these cells.

**\*For correspondence:**
smsmirnakis@bwh.harvard.edu (SS);
poirazi@imbb.forth.gr (PP)

†These authors contributed equally to this work

## Editor's evaluation

This manuscript will be valuable to scientists working on visual neurobiology and cortical processing. It uses a compartmental model to evaluate the relative contribution of basal and apical dendritic trees to the orientation selectivity of layer 2/3 pyramidal cells. There is solid support for the key claims that pertain to the model itself, but there are some questions as to how well the model reflects the biological circuit.

## Introduction

In the primary visual cortex (V1), layer 2 and 3 (L2/3) pyramidal neurons conform to a stereotypical morphology characterized by separate apical and basal dendritic arbors. The apical tree consists of a thick apical trunk that extends into L1 and splits into an apical tuft. The basal tree consists of numerous dendritic segments, the majority of which sprout directly from the base of the soma, indicating that they can directly influence somatic output (*Spruston, 2008*). This compartmentalization of V1 neurons is also evident in the wiring diagram describing each tree: layer 4 (L4) and L2/3 pyramidal neurons synapse with basal dendrites of L2/3 pyramidal neurons, providing them

with feedforward input (*Coogan and Burkhalter, 1990*; *Larkum, 2013*). The apical dendrites of the L2/3 pyramidal neurons instead receive feedback input from higher order areas of the cortex such as the secondary and tertiary visual cortices, lateromedial area and prefrontal cortex (*Coogan and Burkhalter, 1990*; *Larkum, 2013*), as well as orientation-tuned thalamocortical input (*Chen et al., 2013*; *Cruz-Martín et al., 2014*; *Jia et al., 2010*; *Roth et al., 2016*). These anatomical and connectivity features of the apical and basal trees, complemented by their distinct biophysical properties (*Cho et al., 2008*), shape both the local dendritic processing and the integration of the two input streams at the soma.

Few studies have undertaken the technically difficult task of measuring single spine tuning properties in the mouse visual cortex (*Chen et al., 2013*; *Iacaruso et al., 2017*; *Jia et al., 2010*). These studies have shown that single spines of L2/3 pyramidal neurons exhibit orientation selectivity, with synaptic orientation preferences varying even on the same branch (*Iacaruso et al., 2017*; *Jia et al., 2010*). These findings raise the question of how dendritic integration of sparse synaptic inputs along the apical and basal trees shape orientation tuning in L2/3 V1 pyramidal neurons.

The effect of synaptic input on neuronal output depends on the biophysical properties of dendritic branches. Older modeling work has established that blockage of dendritic sodium and NMDA activity leads to impaired or abolished orientation selectivity at the level of the soma (*Archie and Mel, 2000*; *Mel et al., 1998*). It is also known that individual dendritic segments of L2/3 neurons generate nonlinear regenerative responses, termed dendritic spikes, independently of the soma, making their exact contribution to somatic output unclear (*Palmer et al., 2014*; *Smith et al., 2013*). This could indicate that multiple dendritic segments need to be activated in order to generate a spike at the soma. On the other hand, it was recently proposed that strong, sparse inputs to the dendrites are sufficient to drive somatic output (*Goetz et al., 2021*). Interestingly, although dendritic spikes in the apical tuft of L2/3 V1 pyramidal neurons influence orientation selectivity (*Goetz et al., 2021*; *Smith et al., 2013*), this orientation selectivity is robust to ablation of the apical tree (*Park et al., 2019*). Altogether, the above studies highlight the need to scrutinize the functional effect that synaptic inputs along the basal and apical dendritic arbors have on the soma, building towards a realistic theory of visual processing.

In this work, we used a detailed biophysical model of a L2/3 V1 pyramidal cell (*Park et al., 2019*), to tease apart the relative contributions of apical vs. basal dendritic trees in somatic orientation tuning. We start by examining the threshold and strength of dendritic non-linearities, which are found to vary along the apical and basal dendrites. By comparing model output to in vivo two-photon calcium fluorescence imaging data, we find that dendritic and somatic spontaneous activity patterns in the model adequately reproduce experimental data. We then simulate orientation tuning in our model and evaluate its responses to specific combinations of dendritic disparity and synaptic distributions. We find that the tuned output of the model does not always match the linear summation of tuned input, instead deviating towards orientation preferences more closely matching either the apical or basal mean orientation preference.

We continue by evaluating the dependence of somatic spikes on dendritic voltage-gated sodium channel conductances by performing just-in-time interventions that nullify sodium conductance on either the apical or basal tree prior to a somatic spike. We find that the majority of stimulus-driven somatic spiking activity is dependent on sodium spikes generated in the apical tree, suggesting that apical and basal trees contribute via different mechanisms to tuned somatic output. We investigate this further by examining the role of both specific ionic (sodium channels) and synaptic (AMPA/NMDA) mechanisms in shaping orientation tuning. Results indicate that apical sodium channels are critical for proper neuronal tuning, while basal sodium channels do not play as significant a role. Conversely, AMPA and NMDA activity of both the apical and basal trees is required for orientation tuning at the level of the soma. A potential reason for this is that basal dendritic branches feature higher sodium spiking thresholds for similar electrotonic constant-normalized length values (i.e. 'electrotonic length') compared to apical branches, reducing their propensity to generate sodium spikes compared to apical dendrites.

Overall, our simulations suggest a synergistic effect between apical and basal trees, as somatic spikes are only reliably produced when apical dendritic sodium spikes coincide with basal synaptically driven depolarizations and/or spikes. In this manner, apical and basal dendrites contribute to neuronal function using different mechanisms, ionic or synaptic, which together drive action potential generation in this neuronal type.

**Table 1.** Outline of passive, active, and synaptic mechanisms present in the model neuron.

| Compartment type | Passive/active mechanisms | Synaptic mechanisms |
|---|---|---|
| Soma | Hodgkin/Huxley voltage-gated $Na^+$ channels | $GABA_A$ (background-driven) |
| Basal dendrites | Hodgkin/Huxley voltage-gated $K^+$ channels | AMPA (background-driven) |
| | Muscarinic voltage-gated $K^+$ channels | NMDA (background-driven) |
| | A-Type voltage-gated $K^+$ channels | $GABA_A$ (background-driven) |
| | T-Type $Ca^{++}$ channels | AMPA (stimulus-driven) |
| | High voltage activated (HVA) $Ca^{++}$ channels | NMDA (stimulus-driven) |
| | Calcium-dependent $K^+$ channels | $GABA_A$ (stimulus-driven) |
| Apical dendrites | Active ATP $Ca^{++}$ pumps | |

## Methods
### Model description
The single neuron model is based on the L2/3 V1 pyramidal cell model of *Park et al., 2019* created in the NEURON simulation environment (*Hines and Carnevale, 2001*). As the model makes use of a variety of passive and active mechanisms ( *Tables 1–4*), its electrophysiological properties were validated against experimental data (*Park et al., 2019*). Neither the morphology nor the biophysical properties of the source model were altered, barring specific ion channel or synaptic mechanism conductance changes wherever specified (see Extended methods for details). Apical stimulus-driven synaptic inputs have had a 10ms delay added, to better reflect experimental data on feedback synaptic inputs (*Ju et al., 2020*). Simulations with synchronous activation of the apical and basal synaptic inputs resulted in qualitatively similar results (data not shown). For a more detailed description of the model, see *Park et al., 2019* and Extended methods, Model description.

### Simulation information
All simulations were performed on our High-Performance Computing Cluster (Rocks 7.0) with 624 cores and 3.328 TB of shared RAM under CentOS 7.4 OS (*Papadopoulos et al., 2003*), through the NEURON simulation environment (*Hines and Carnevale, 2001*). Model neuron output measurements were obtained at a sampling rate of 10 KHz (dt = 0.1ms). Unless specified otherwise, voltage recordings from any section of the model (soma / dendrites) were performed at the midpoint of the respective section. Simulation duration was 2500 ms, with stimulus onset (where present) at t = 500 ms. For information on manipulations of biophysical properties for specific experiments, see Extended methods, Biophysical manipulations. For details on specific simulation protocols, see Extended methods, Simulation protocols.

### Two-photon imaging
All experimental protocols were approved by Brigham and Women's Hospital (BWH) Institution Animal Care and Use Committee. Male and female wild-type (C57BL/6) mice were purchased from

**Table 2.** Outline of membrane mechanism conductances (not synaptic).
Reproduced from *Park et al., 2019*.

| Conductance (mS/cm²) | Soma | Apical | Basal |
|---|---|---|---|
| $g_{Na}$ | 0.505 | 0.303 | 0.303 |
| $g_{Kdr}$ | 0.05 | $1.5*10^{-3}$ | $1.5*10^{-3}$ |
| $g_{Km}$ | $2.8*10^{-3}$ | $1.27*10^{-3}$ | $1.27*10^{-3}$ |
| $g_A$ | 5.4 | Diameter ≤0.8 μm: 108<br>Diameter >0.8 μm: 10.8 | Diameter ≤0.8 μm: 108<br>Diameter >0.8 μm: 10.8 |
| $g_T$ | 0.03 | x≤260 μm: $0.029*\sin(0.009*x+0.88)$<br>x>260 μm: 0.012 | $0.03+6*10^{-5}*x$ |
| $g_{HVA}$ | $0.05*10^{-3}$ | x≤260 μm:<br>$0.049*10^{-3}*\sin(0.009*x+0.88)$<br>x>260 μm: $0.02*10^{-3}$ | $0.05*10^{-3}+10^{-7}*x$ |
| $g_{KCa}$ | $2.1*10^{-3}$ | $2.1*10^{-3}$ | $2.1*10^{-3}$ |

**Table 3.** Outline of model electrophysiological properties.
RMP: resting membrane potential, IR: Input Resistance measured at hyperpolarizing current (–0.04 nA), AP: action potential, AHP: after hyperpolarization measured at depolarizing current (0.16 nA), P-T peak-trough. Reproduced from *Park et al., 2019*.

|  | Model | Cho et al., 2010 |
|---|---|---|
| RMP, mV | –79 | –78.56 ± 1.34 |
| IR, MΩ | 123.6 | 125.2 ± 8.2 |
| τ, ms | 17.3 | 16 ± 0.7 |
| AP amplitude, mV | 66.1 | 67.8 ± 1.8 |
| AP threshold, mV | –41.8 | –37.7 ± 1.3 |
| AHP, mV | 17.9 | 13.3 ± 0.5 |
| P-T time, ms | 38.6 | 55.3 ± 2.7 |
| AP adaptation | 1.16 | 1.18 ± 0.02 |

The Jackson Laboratory and bred for experiments. A total of 2 mice were used in these experiments, with recordings from 11 neurons used for data analysis. For details on chronic window implantation, sparse labeling and in vivo imaging methodology, see Extended methods, Two-photon microscopy.

## Data analysis

Data analysis of both modeling and two-photon imaging experiments was performed through Python 3.8+, using publicly available libraries as well as custom code. Analysis procedures include methods for dendritic spike detection, quantification of dendritic non-linearities, calculation of dendritic electrotonic length, analysis of two-photon imaging recordings, processing of calcium fluorescence traces, detection of calcium events, conversion of voltage traces into approximate calcium fluorescence traces, and comparison of in vivo fluorescence data with converted model voltage traces. For specifics, see Extended methods, Data analysis.

## Results
### The degree of non-linearity varies across basal and apical model dendrites

To ensure the validity of our modeling approach, we first examined dendritic integration by characterizing the sodium spike properties of all dendritic segments (*Figure 1A*), wherein a specific number of synapses is synchronously activated by two pulses at 50 Hz while all other compartments are modeled as passive ('I3P'; see Extended methods, Iterative paired-pulse protocol (I3P)), and the response of each dendritic segment is assessed. Results show that the model can produce realistic dendritic spikes (*Figure 1—figure supplement 1C, D*), while the properties of the spiking events vary between dendritic segments (*Figure 1B–D*, *Figure 1—figure supplement 1A, B*). Both apical and basal dendritic segments exhibit a large degree of variation in their spiking thresholds (*Figure 1E, F*; threshold synapse count mean and standard deviations: basal 12.286±6.649 sodium, 51.429 ± 13.179 NMDA; apical 14.442 ± 13.825 sodium, 40.721 ± 23.675 NMDA). Dendritic segments with high sodium spiking thresholds also tended to exhibit higher NMDA spiking thresholds. To quantitatively

**Table 4.** Outline of synaptic mechanism conductances and time constants.
Reproduced from *Park et al., 2019*.

|  | Conductance (nS) | $\tau_1$, ms | $\tau_2$, ms |
|---|---|---|---|
| NMDA | 1.15 | 2 | 30 |
| AMPA | 0.84 | 0.1 | 2.5 |
| GABA$_A$ | 1.25 | 0.2 | 1.4 |

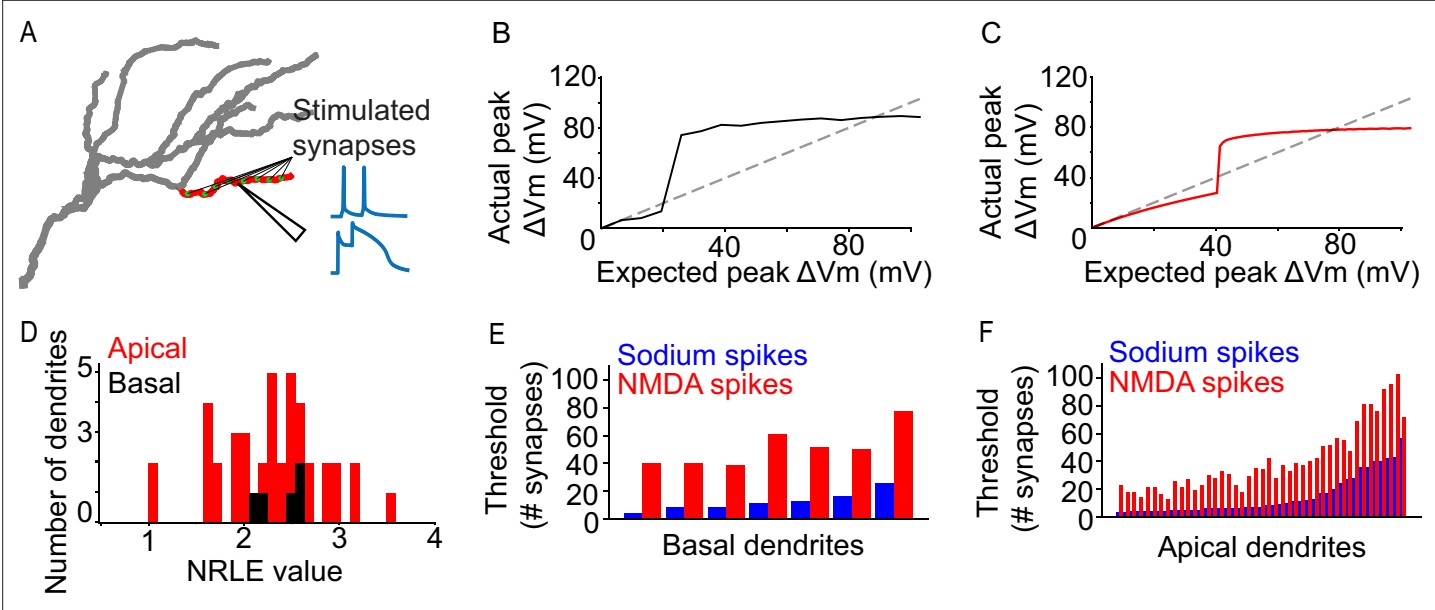

**Figure 1.** Dendritic properties of a L2/3 V1 neuron model. (**A**) Graphical representation of the iterative paired-pulse protocol. Green dots represent allocated synapses. Synapses were activated only within the designated (red) dendritic segment, with all other segments being passive. We record the voltage response of the dendritic segment with NMDA channels disabled or enabled, acquiring information on dendritic sodium and NMDA spikes, respectively. (**B**) Example of an 'expected vs. actual plot', showing the response produced by a single basal dendritic segment (solid black line) as a result of voltage-gated sodium channel activity elicited through simultaneous activation of 1–200 AMPA synapses. The dashed black line indicates the expected dendritic response assuming a linear dendrite. (**C**) Same as B, but for an apical dendritic segment. (**D**) NRLE value histogram for all apical (red) and basal (black) dendritic segments. (**E**) Bar plot of per-dendrite nonlinearity thresholds for all basal dendritic segments, sorted by the sodium spike thresholds. Note that NMDA spike thresholds largely follow the sodium spike threshold increase. (**F**) As E, for all apical dendritic segments.

The online version of this article includes the following figure supplement(s) for figure 1:

**Figure supplement 1.** Visualization of dendritic non-linear properties.

characterize the nonlinear behavior of each dendritic segment we used the Nonlinearity Relative to Linear Extrapolation (NRLE) metric (*Behabadi et al., 2012*), which is an estimate of the degree of non-linear dendritic integration within a specific branch (see Extended methods, Quantification of dendritic nonlinearities). Results indicate that all dendritic segments of the model neuron exhibit non-linear input-output functions (*Figure 1D*, NRLE = 1 denotes linear, NRLE <1 sublinear, and NRLE >1 supralinear), in line with experimental observations (*Smith et al., 2013*; *Wilson et al., 2016*). In addition, the NRLE values are relatively widely dispersed (values ranging from >1 to~3.5), providing a large dynamic range of possible dendrite-level computations. The above results show that there is diversity in apical and basal dendritic integration that can be exploited to modulate orientation tuning. The above differences are, at least in part, due to the morphological and electrophysiological features of dendritic segments, as we found that the sodium spiking propensity varies as a function of dendritic segment volume and electrotonic length (see Figure 5).

## Model neuron features robust orientation tuning

Given the diverse integration properties of basal and apical dendrites in our model cell, we sought to assess their functional cooperation when receiving a more realistic set of background- and stimulus-driven inputs. Stimulus driven synapses were modeled with individual orientation preferences (*Jia et al., 2010*; *Park et al., 2019*), leading them to exhibit different firing rates depending on the orientation of the presented stimulus (for details, see Extended methods, Model description). As dictated by experimental data (*Chen et al., 2013*), background-driven synapses comprise the majority of inputs to the model, although they are activated at lower rates than stimulus-driven synapses (see Extended methods, Model description). To validate their impact on somatic output, we compared the simulated spontaneous activity with in vivo calcium two-photon fluorescence imaging data (see Extended methods, Analysis and comparison of true vs "generated" fluorescence data and *Figure 2—figure*

*supplements 1–4*). Although this approach has its limits, it allows us to use two-photon imaging data to increase model reliability. We found that spontaneous activity patterns in the model are very similar to those seen in two-photon calcium imaging data (*Figure 2—figure supplement 1E, F*).

Next, we combined stimulus-driven and background-driven activity to derive the orientation tuning curve of the model neuron (see Extended methods, Orientation tuning validation protocol). We focused on the biologically plausible model (see Extended methods, Model description), whereby the distribution of tuned synaptic inputs onto the basal and apical trees is in line with experimentally reported values (60% vs 40%, respectively; *Defelipe and Fariñas, 1992*). The resulting orientation tuning curve (preferred orientation firing rate: 1.55 ± 0.3 Hz, orthogonal orientation firing rate: 0.21 ± 0.12 Hz, OSI 0.77 ± 0.11, tuning width 44 ± 9.17°; *Figure 2A*) indicates that the model neuron is well-tuned (OSI >0.2, tuning width <80°; see Extended methods, Orientation tuning validation protocol), in agreement with our prior work using the original version of this model (*Park et al., 2019*). We also reproduced the apical tree ablation experiment from that publication, finding that orientation tuning does indeed persist post-ablation (*Figure 2—figure supplement 5*; see Extended methods, Ablation protocol).

Having established the orientation tuning of our model cell, we tried to disentangle the effects of the apical and basal trees by introducing dendritic disparity (*Figure 2B*; see Extended methods, Orientation disparity protocol). We used the disparity protocol, in which the apical and basal trees are tuned to different orientations: the apical tree features synaptic orientation preferences sampled from a Gaussian distribution with a mean of 0°, whereas the basal tree uses the same method, but with a different mean (0° - 90°, in steps of 10°). The expected orientation preference of the model neuron is calculated using Euler's Formula, which assumes that synapses contribute to overall orientation preference in a linear fashion (for details, see Extended methods, Orientation disparity protocol). We found that the model neuron diverges from the expected orientation preference (linear summation of synaptic inputs) and exhibits a bias towards the preferred orientation of the basal tree. This is an example of *basal dominance* (*Figure 2D*) and was previously observed using the original version of the model used here (*Park et al., 2019*). Of note, the model neuron did not exhibit adequate tuning (i.e. 20% or more of neurons featured OSI <0.2 or tuning width >80°; see Extended methods, Orientation tuning validation protocol) beyond a dendritic disparity of 50°.

To assess whether this result depends on the particular synaptic distribution used, we repeated the experiment using two different synaptic distribution profiles – the 'inverse biologically plausible' (40% on the basal tree, 60% on the apical tree) and 'even' (50% on both trees) models (see Extended methods, Model description). We found that, indeed, the deviation from expectation displayed by the neuron appears to be decreased in the 'even' model, and is clearly reversed in the 'inverse biologically plausible' model. Similar to the 'biologically plausible' model, these two models did not feature adequate tuning past 50° of dendritic disparity. Overall, the observed deviation from expectation indicates that apical and basal synaptic inputs are not linearly combined, but rather interact in a non-linear manner to produce somatic output. This interaction is influenced by the synaptic distribution along the two dendritic trees, as well as their mean synaptic orientation preferences.

## Apical sodium spikes are the primary driver of neuronal output

The previous results show that orientation tuning emerges from non-linear interactions between apical and basal trees. To elucidate the contribution of each tree in orientation tuning, we selectively interrupt specific aspects of their operation and assess the effect of the intervention on somatic tuning. We start by selectively and precisely modifying the sodium spiking activity of each dendritic tree, while controlling for the type of input received (stimulus-driven vs background-driven). Specifically, we use the ionic intervention protocol (see Extended methods, Ionic intervention protocol) to remove the sodium-induced depolarization that is provided by either the apical or the basal tree, just before the generation of a somatic spike.

We start by using a control version of the 'biologically plausible' model ('Control background') and a preferred (0°) stimulus (case #1). In this condition, both stimulus-driven and background-driven synapses are active, and most synapses (60%; both stimulus-driven and background-driven) are allocated on the basal tree. For every somatic spike, we perform the intervention protocol and categorize the spike as either: 'apically driven' (dependent on the presence of apical sodium conductances), 'basally driven' (dependent on the presence of basal sodium conductances), or 'cooperative'

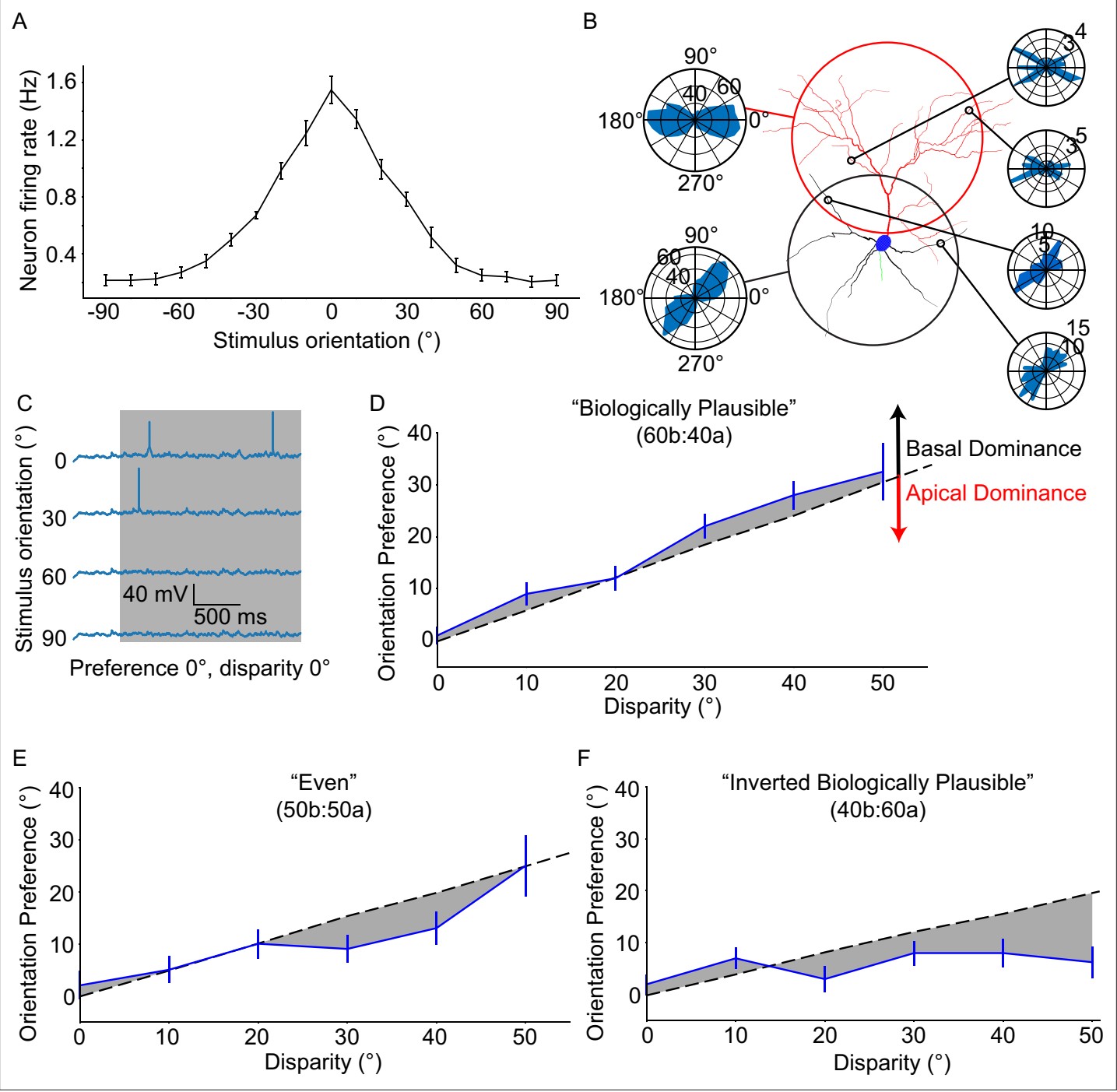

**Figure 2.** Neuronal orientation preference is robust and resists dendritic tuning disparity. (**A**) Orientation tuning curve for the 'biologically plausible' model. (**B**) Example from a configuration of a model neuron set to a disparity of 40°. Right-side polar plots display the distribution of synaptic orientation preferences for the indicated dendritic segments. Left-side polar plots display the distribution of synaptic orientation preferences for the apical (top) and basal (bottom) dendritic trees. (**C**) Example voltage traces recorded at the soma of a model neuron from (**A**) Note the clearly different responses between the preferred orientation (0°) and all others. (**D**) Orientation preference plot for the 'biologically plausible' model featuring various degrees of disparity. Grey dashed line denotes the expected orientation preference per degree of disparity. Black and red arrows point in the direction of basal and apical dominance, respectively. (**E**) As D, but for the 'even' model. Note that dominance has shifted from the basal domain towards the apical domain. (**F**) As D, but for the 'inverted biologically plausible' model. This configuration clearly features apical dominance. Error bars: Standard error of the mean, for all panels.

The online version of this article includes the following figure supplement(s) for figure 2:

*Figure 2 continued on next page*

*Figure 2 continued*

**Figure supplement 1.** Comparing model output with two-photon imaging data.

**Figure supplement 2.** Properties of the GcaMP6s kernel.

**Figure supplement 3.** Waveforms of detected calcium fluorescence events.

**Figure supplement 4.** Event-triggered averages of detected events and statistical testing of event pairs.

(**A**) Event-triggered averages for somatic (blue) and basal dendrite (red) events in mouse 1. (**B**) Event-triggered averages for somatic (blue) and basal dendrite (red) events in mouse 2. Note the difference in fluorescence intensity compared to mouse 1. Error bars: Standard error of the mean. (**C**) Comparison of the true (two-photon) data distribution (blue) with the null distribution (light red). Distributions are different (K.S. two-sample test; test value ~0.105, p<<0.05). Note that the middle bar for the true data has been truncated (actual value 0.12267).

**Figure supplement 5.** Comparison of orientation tuning behavior seen in the model under control (blue) and apical dendrite ablation (red) configurations.

(dependent on the presence of both apical and basal sodium conductances; *Figure 3A*). We find that the majority of somatic spikes are dependent on apical sodium conductance (N=131 spikes; apically driven 76.51 ± 10.21%, basally driven 9.61 ± 5.03%, cooperative 13.89 ± 11.22%; *Figure 3B*), which is surprising, considering the proximity of the basal dendrites to the soma, as well as the salient nature of the feedforward, visually evoked inputs received by the basal tree. These somatic events are a mix of stimulus-driven and background-driven input. In order to separate the two, we changed the orientation of the stimulus being presented to 90° (i.e. orthogonal to the orientation preference of the neuron), rendering the vast majority of stimulus-driven synapses inactive, while keeping background-driven synapses active as normal (case #2). We found that the resulting percentages are similar to the previous condition (N=25 spikes; apically driven 71.07 ± 31.56%, basally driven 17.5 ± 22.5%, cooperative 1.43 ± 4.29%; *Figure 3D*), indicating that the dependence of somatic spiking activity on dendritic sodium conductance does not significantly differ between instances of stimulus-driven and background-driven activity.

However, it is possible that background-driven synapses are actually the ones that determine somatic spiking. While firing at low rates (0.11 Hz vs 0.3 Hz for stimulus-driven; see Extended methods, Model description), these synapses are vastly more numerous compared to stimulus-driven synapses (75% vs 25% of total synapses). To test this possibility, we equalized the contribution of background-driven synapses to apical and basal tree outputs, by introducing a new condition ('Evenly distributed background') whereby the model is configured as in case #1 (i.e. 'biologically plausible' model), but the distribution of *background-driven synapses* is the same on both trees (i.e. excitatory: 50% apical, 50% basal; inhibitory: 46.5% apical, 46.5% basal, 7% soma). In this way, we ensure that the greater proportion of background-driven synapses on the basal tree is no longer a confounding factor. Presenting a stimulus of the preferred orientation (0°) under these conditions (case #3), we find that the resulting percentages (N=156 spikes; apically driven 85.08 ± 7.34%, basally driven 5.15 ± 6.72%, cooperative 9.78 ± 8.17%; *Figure 3C*) are similar to those seen in case #1, indicating that the distribution of background-driven synapses does not significantly affect somatic spike generation.

By changing the orientation of the presented stimulus to 90° for this altered model configuration, we can also evaluate whether the same conclusion holds for background-driven spikes (case #4). Indeed, the results (N=42 spikes; apically driven 85.83 ± 29.83%, basally driven 2.5 ± 7.5%, cooperative 1.67 ± 5.0%; *Figure 3E*) are not significantly different from those seen in case #2. These simulations predict that somatic spike generation reliably depends on apical sodium conductance, a seemingly counterintuitive finding, as feedforward stimulus-driven input reaches mostly the basal tree and not the apical tree.

To gain more insight into these findings, we looked at the spatiotemporal evolution of voltage through different apical and basal paths during the occurrence of an apically or basally driven spike (*Figure 3—figure supplement 1*). For heavily branching apical dendrites, input summation and/or voltage propagation in one segment along the path exceeds the sodium spiking threshold, leading to a cascade of dendritic sodium spikes both in the direction of the soma as well as towards any terminal dendrites that did not yet generate sodium spikes of their own (e.g. *Figure 3—figure supplement 1H*; dendritic sodium spiking does not originate from the terminal segment, but propagates to it). The lack of extensive branching on the basal dendrites, on the other hand, renders them less predictable in terms of sodium spiking activity (e.g. *Figure 3—figure supplement 1C*). Although the intervention

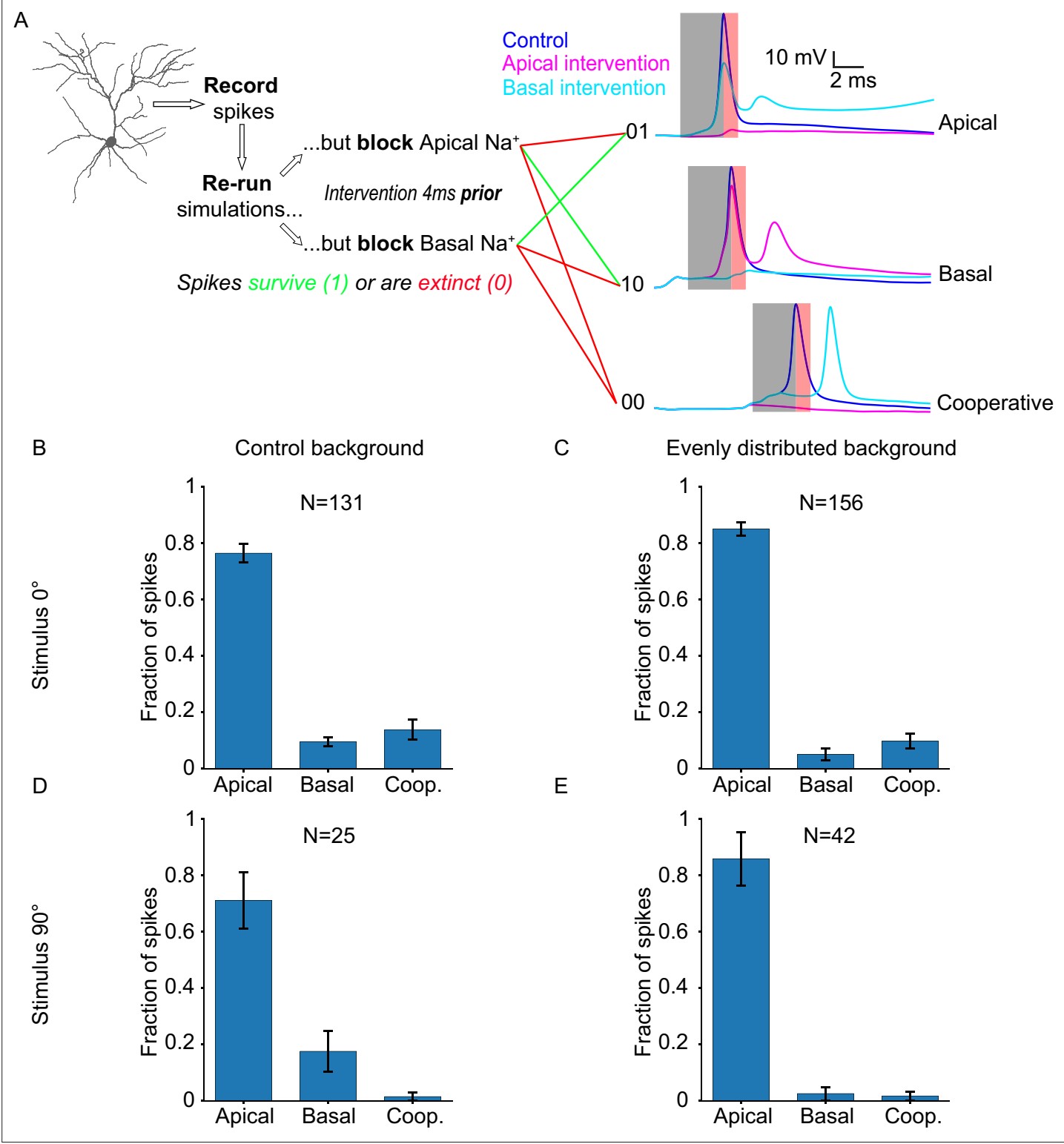

**Figure 3.** Neuronal output relies disproportionately on apical sodium conductance. (**A**) Diagram describing the ionic intervention protocol, with example traces on the right. Shaded areas on the traces denote the pre-spike (grey) and post-spike (red) intervention time windows. (**B**) Somatic spike dependence on dendritic sodium conductance for the 'biologically plausible' model with a stimulus of 0° (preferred orientation). (**C**) Same as B, but background synapses are evenly distributed across the apical and basal dendritic trees, while stimulus-driven synapses follow the 'biologically plausible' (60% basal, 40% apical) model. (**D**) Somatic spike dependence on dendritic sodium conductance for the 'biologically plausible' model with a stimulus of 90° (orthogonal/non-preferred orientation). (**E**) Same as D, but background synapses follow the 'even' model (evenly distributed), while stimulus-driven

*Figure 3 continued*

synapses follow the 'biologically plausible' (60% basal, 40% apical) model. 'N' is the total number of spikes for each condition. Error bars: Standard error of the mean, for all panels.

The online version of this article includes the following figure supplement(s) for figure 3:

**Figure supplement 1.** Example traces showing the spatiotemporal evolution of somatodendritic voltage before, during and after the generation of a dendritic-sodium-mediated somatic spike, across different dendritic paths.

experiments are required in order to decisively determine the origin of each somatic spike, the temporal precedence of dendritic sodium spiking reliably indicates which tree drives somatic spiking activity (e.g. compare *Figure 3—figure supplement 1B, C, E, F*).

## Apical and basal dendrites influence somatic tuning via different mechanisms

The previous results are counter-intuitive: although we would expect both apical and basal dendritic segments to contribute to somatic spiking, the contribution of the apical tree overshadows that of the basal tree, at least with respect to the role of sodium conductances. However, as indicated by our results in *Figure 2D*, the basal tree, being the bearer of the majority of feedforward inputs, has a dominant role in orientation tuning. Thus, we assume that the basal tree exerts its influence via a different mechanism. To assess the veracity of this assumption, we performed a series of experiments aimed at evaluating the impact of ionic (see Extended methods, Sodium (channel) blockage and Ionic intervention protocol) and synaptic (see Extended methods, Synaptic modulation, Input manipulation and Synaptic intervention protocol) mechanisms on neuronal orientation tuning, using the biologically plausible model (see Extended methods, Stimulation protocol).

First, we assess the tuning characteristics of the neuron when stimulus-driven input is removed from either the apical or the basal tree (see Extended methods, sectionInput manipulation), and compare it with the control case. We find that the neuron remains tuned in all cases, albeit with significantly decreased OSI values (dependent two-sample t-test, control vs basal: $p \approx 0.000037$; control vs apical: $p \approx 0.000007$) (*Figure 4A*). Removal of apical stimulus-driven input has a much smaller effect on tuning than removal of basal input (dependent two-sample t-test, apical vs basal: $p \approx 0.0095$), which is in line with our previous findings (*Park et al., 2019*).

Next, we use the synaptic intervention protocol to selectively reduce synaptic (AMPA and NMDA) weights by 50% on either the apical or basal tree for a brief time only, and only when a somatic spike is about to occur (see Extended methods, seSynaptic intervention protocol). We find that somatic tuning is completely abolished when manipulating the basal but not the apical synaptic weights, although tuning is significantly affected by loss of apical inputs as well (*Figure 4B*; dependent two-sample t-test, control vs apical: $p \approx 0.00065$). These results indicate that synaptic input from the basal tree is critical for orientation tuning, while apical tree synaptic inputs play a less important role.

Finally, we assess the impact of sodium conductance nullification (as in *Figure 3*), this time on orientation tuning rather than somatic spiking. In line with our previous results (*Figure 3*), we find that sodium conductance nullification in the apical tree completely abolishes orientation tuning (*Figure 4C*) while the same manipulation in the basal tree has no impact on tuning (dependent two-sample t-test, control vs basal: $p \approx 0.24125$).

## Sensitivity analysis

To assess whether the above findings are robust to biologically relevant variability in the conductance values of sodium channels, we performed a sensitivity analysis whereby we reduced the sodium conductance only in the apical tree by either 5% or 10% (*Figure 4—figure supplement 1*). For a decrease of 5%, we found that elimination of the apical but not the basal sodium component results in total loss of tuning (dependent two-sample t-test, control vs basal p-value: 0.32549). Importantly, in this scenario the apical tree remains the primary driver of somatic activity as the majority of somatic spikes are apically driven (*Figure 4—figure supplement 1A* N=345 spikes; 60 ± 15% of spikes are apically driven, 30 ± 18% are basally driven, and 10 ± 8% are cooperatively driven, for details on the characterization of spiking activity, see Results, Apical sodium spikes are the primary driver of neuronal output). However, a 10% decrease in the apical sodium conductance results in total

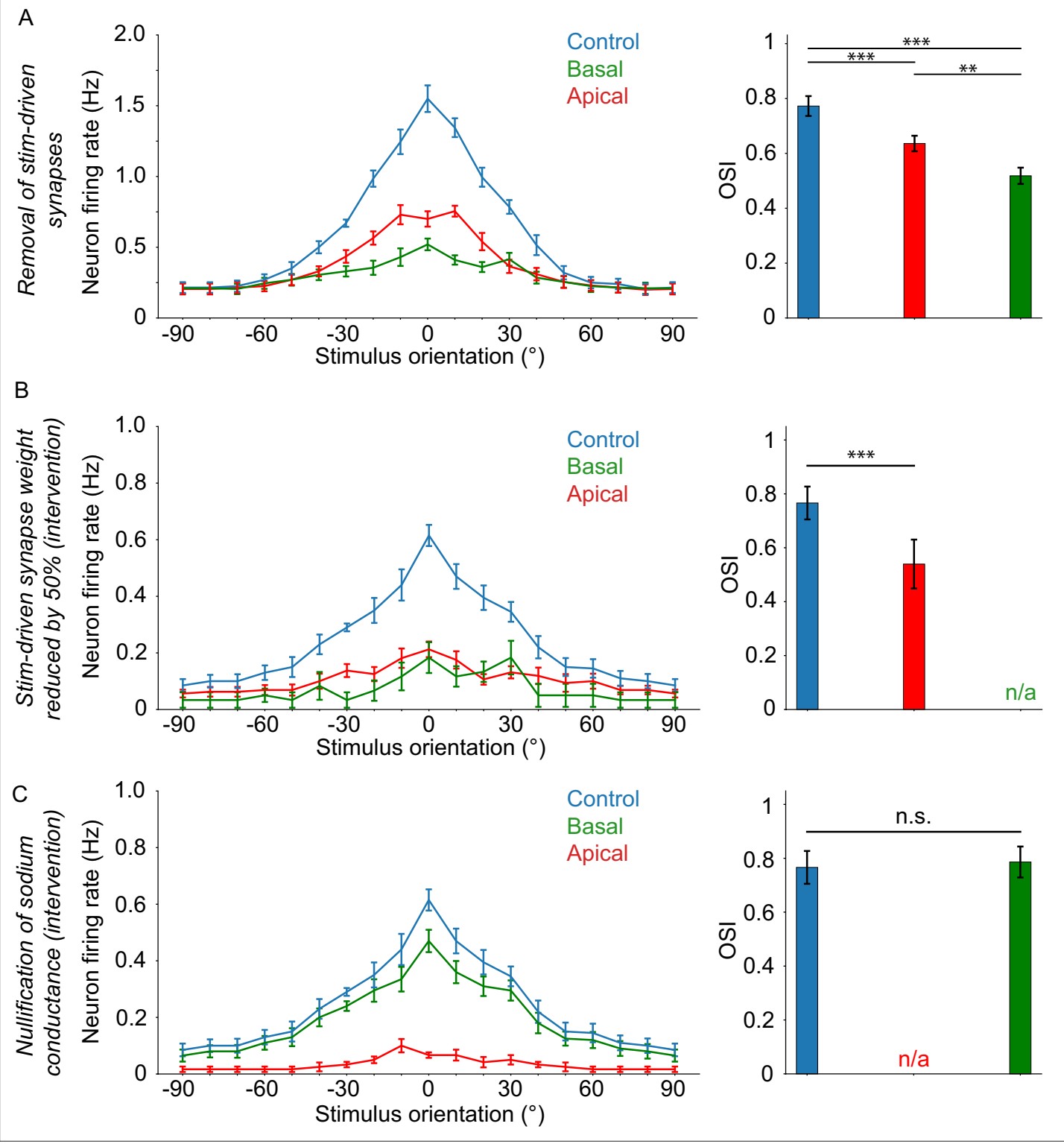

**Figure 4.** Orientation tuning critically depends on apical sodium conductance and ampa/nmda activity. (**A**) Orientation tuning curves (left) and mean OSI values (right) for the experiment where stimulus-driven synapses are completely absent on either the apical ('apical target'; red) or basal ('basal target'; green) tree. (**B**) Selective reduction of stim-driven synaptic (AMPA/NMDA) weights by 50% 30ms prior to somatic spike occurrence (lasting up to 10ms after the pre-recorded somatic spike timing), on either the apical (apical target; red) or basal (basal target; green) tree. (**C**) Nullification of sodium conductances 3ms prior to somatic spike occurrence (lasting up to 1ms after the pre-recorded somatic spike timing), on either the apical (apical target; red) or basal (basal target; green) tree. Error bars: Standard error of the mean, for all panels. If present, 'n/a' denotes a lack of orientation tuning for that

*Figure 4 continued on next page*

*Figure 4 continued*

specific experimental setup. Stars denote statistical significance at a significance level (α) of 0.05 (*), 0.01 (**) or 0.001 (***), whereas 'n.s.' stands for 'no statistically significant difference', that is p-value ≥ α. Dependent two-sample t-tests were used in all cases.

The online version of this article includes the following figure supplement(s) for figure 4:

**Figure supplement 1.** Results from experiments similar to *Figure 4C*, only with a reduction in apical sodium conductance.

loss of tuning when either the apical or the basal sodium component is eliminated (*Figure 4—figure supplement 1B*). In this case, a large number of spikes are completely lost (N=190 spikes vs 345), and out of the remainder, 41 ± 25% are apically driven, 48 ± 28% are basally driven, and 11 ± 12% are cooperatively driven. We further compare these two conditions (5% and 10% reduction) with one where the apical sodium conductance is unchanged (*Figure 4—figure supplement 1C* identical to *Figure 4C*). Similarly to the results seen in the 5% reduction, elimination of the apical – but not the basal – sodium component results in total loss of tuning (dependent two-sample t-test, control vs basal p-value: 0.24125). In this case, 80 ± 9% of spikes are apically driven, 13 ± 8% are basally driven, and 7 ± 4% of spikes are cooperatively driven. Notably, the number of spikes is drastically higher in this condition (N=885 spikes in the unchanged condition vs. N=345 for the 5% reduction and N=190 for the 10% reduction).

The above simulations show that, as expected, the reduction in apical sodium conductance results in a decrease of the overall spiking rate of the neuron (number of spikes: 885 in the unchanged condition, 345 for the 5% reduction, and 190 for the 10% reduction). Even so, decreasing apical sodium conductance by 5% failed to negate its role in somatic spike generation, with the apical tree remaining the primary contributor (60 ± 15% of spikes were apically driven vs 30 ± 18% basally driven; *Figure 4—figure supplement 1A*). A decrease of 10%, however, exerts greater influence as an ever-larger number of apically driven spikes are lost, shifting the percentages to a more even distribution between apically driven and basally driven (41 ± 25% vs 48 ± 28%, respectively; *Figure 4—figure supplement 1B*). Of note, in this particular case the neuronal firing rate is well below the experimentally relevant range, suggesting that such a reduction is not physiological (see *Figure 4—figure supplement 1A, B*). Overall, our sensitivity analysis suggests a greater-than-expected contribution of apical sodium non-linearities in somatic spike generation. This finding is in line with experimental work showing that L2/3 V1 neurons can produce spikes even in the absence of feedforward (i.e. basal stimulus-driven) activity (*Keller et al., 2020*).

Overall, our simulations predict that apical sodium channels are necessary for both spike generation in general (*Figure 3*) as well as for orientation tuning in particular (*Figure 4C*). Additionally, AMPA and NMDA activity is necessary for both apical and basal trees (*Figure 4A and B*). This means that the apical tree most likely exerts its influence on somatic spiking via sodium-mediated spiking, whereas the basal tree mostly utilizes AMPA- and NMDA-driven depolarizations, rather than sodium spikes.

## Tree-specific morphological and electrophysiological properties influence dendritic and somatic behavior

Our simulations predict that apical and basal dendritic segments exert causal influence on somatic spike generation via separate mechanisms. In an attempt to explain these differences in contribution, we looked at the anatomical and electrophysiological properties of the two types of dendrites (apical/basal). Specifically, we searched for links between anatomical/electrophysiological properties of dendrites and (a) signal attenuation or (b) dendritic excitability (number of synapses required to drive a local sodium spike, or 'threshold'; see also *Extended methods,* Quantification of dendritic impact on somatic output for details).

To ensure a fair comparison among different dendritic branches, we induced a 20 mV local EPSP (actual amplitudes: 19.78±0.98 mV) and assessed its attenuation at the soma (see Extended methods, Quantification of dendritic impact on somatic output). We found that EPSP attenuation is greater in apical versus basal dendritic segments of similar length (*Figure 5A*; p-value 0.000002), diameter (*Figure 5B*; p-value 0.005864), or volume (data not shown; p-value 0.000001). When normalized by distance from the soma (*data not shown*; p-value 0.42941), dendritic path volume (data not shown; p-value 0.027934), or electrotonic length (data not shown; p-value 0.666959), EPSP attenuation did not significantly differ between apical and basal dendritic segments. We also examined the relationship

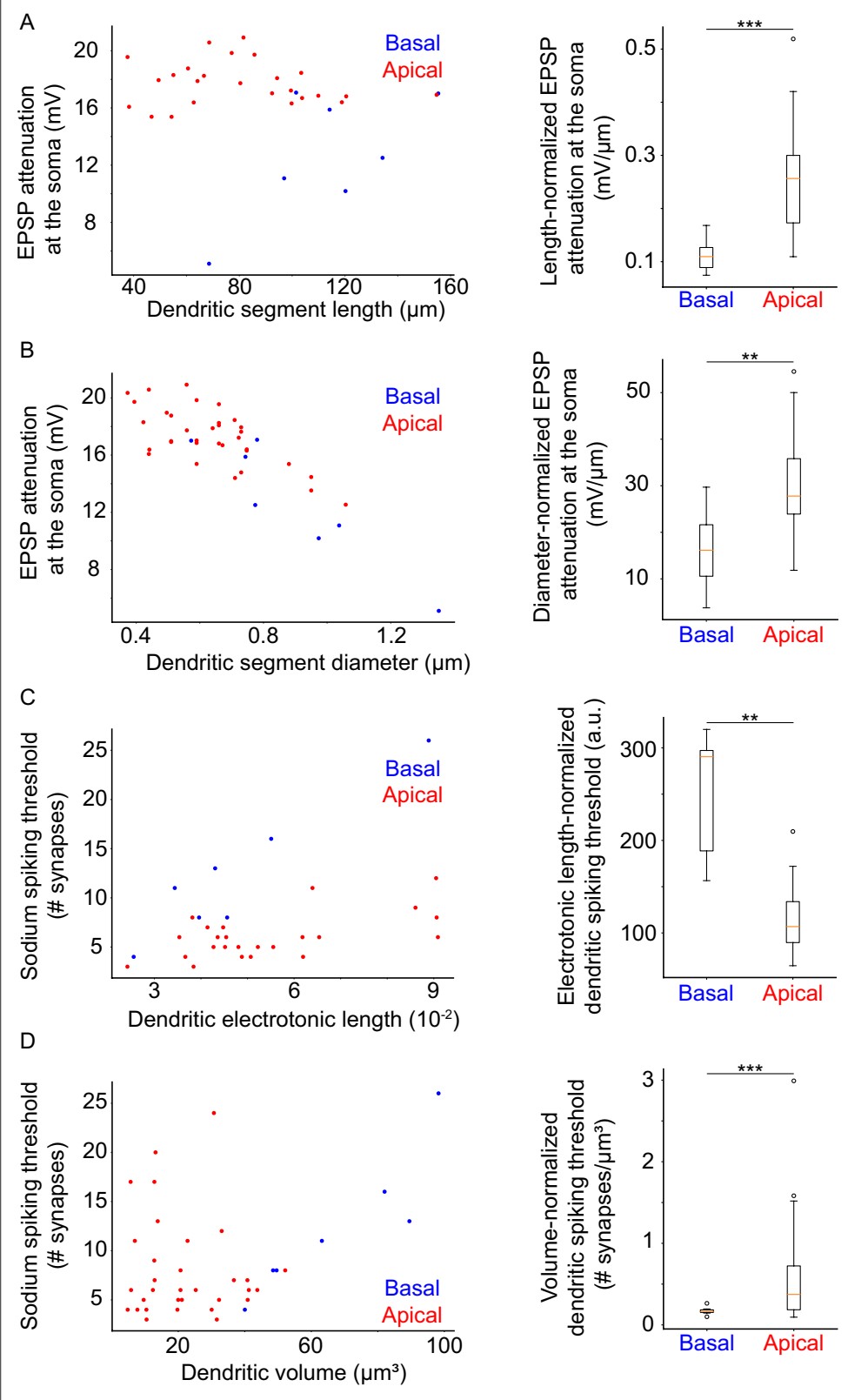

**Figure 5.** Dendritic morphological and electrophysiological properties affect both local and somatic behavior. (**A**) Left: Attenuation of a 20 mV dendritic EPSP (measured at the soma) as a function of dendritic segment length. Right: Comparison of length-normalized EPSP attenuation at the soma between apical and basal dendritic segments. Apical dendritic segments feature greater attenuation. (**B**) Left: Attenuation of a 20 mV dendritic EPSP

*Figure 5 continued on next page*

*Figure 5 continued*

(measured at the soma) as a function of dendritic segment diameter. Right: Comparison of diameter-normalized EPSP attenuation at the soma between apical and basal dendritic segments. Apical dendritic segments feature greater attenuation. (**C**) Left: Dendritic sodium spiking thresholds as a function of dendritic electrotonic length. Right: Comparison of electrotonic length-normalized sodium spiking thresholds between apical and basal dendritic segments. Apical dendritic segments feature lower thresholds. (**D**) Left: Dendritic sodium spiking thresholds as a function of dendritic volume. Right: Comparison of volume-normalized sodium spiking thresholds between apical and basal dendritic segments. Apical dendritic segments feature greater thresholds, but also much greater variability of thresholds. Stars denote statistical significance at a significance level ($\alpha$) of 0.05 (*), 0.01 (**) or 0.001 (***), whereas 'n.s.' stands for 'no statistically significant difference', i.e. p-value $\geq \alpha$.

The online version of this article includes the following figure supplement(s) for figure 5:

**Figure supplement 1.** Results from analyses similar to *Figure 5*.

between apical vs. basal EPSP attenuation with respect to high apical dendrite branching order (>3), as well as high apical dendrite distance from the soma (>182.7 µm). We found no significant difference between apical and basal dendrite EPSP attenuation with respect to high apical dendrite branching order (*Figure 5—figure supplement 1D*; apical attenuation: 16.59±3.18 mV, basal attenuation: 12.69±4.04 mV, p-value 0.0580). A statistically significant difference did emerge in the case of EPSP attenuation in basal dendrites versus distant (d≥182.7 µm) apical dendrites (*Figure 5—figure supplement 1C*; apical attenuation: 17.64±1.84 mV, basal attenuation: 12.69±4.04 mV, p-value 0.0236). We observed, however, that a particular basal dendritic segment – segment #3, the only one featuring branching in the basal tree – yielded unusually low attenuation values (all basal dendritic segments: 12.69±4.04 mV, outlier: 5.11 mV, basal dendritic segments excluding outlier: 13.96±2.8 mV), and repeated the analysis without this atypical data point, still yielding statistically significant differences between distant apical and basal dendrites (apical attenuation: 17.64±1.84 mV, basal attenuation: 13.96±2.8 mV, p-value 0.0312; see *Extended Methods,* Relationship of branch order and distance with signal attenuation for details). We also repeated the analysis for apical dendrites of high branching order without this outlier, finding no significant changes in the results (apical attenuation: 16.59±3.18 mV, basal attenuation: 13.96±2.8 mV, p-value 0.0956).

We also performed analyses for dendritic sodium spikes (*Figure 5—figure supplement 1*) and found similar results: spike attenuation is greater in apical versus basal dendritic segments, when comparing apical and basal dendritic segments of approximately equivalent length (*Figure 5—figure supplement 1B*; p-value 0.000097), diameter (*Figure 5—figure supplement 1A*; p-value 0.00646), or volume (data not shown; p-value 0.000002). There was no significant difference in spike attenuation between apical and basal dendritic segments when controlling for distance from the soma (data not shown; p-value 0.768547), dendritic path volume (data not shown; p-value 0.061389), or electrotonic length (data not shown; p-value 0.396982). As above, we also examined dendritic spike attenuation at the soma with respect to high apical dendrite branching order and high apical dendrite distance from the soma. There was no significant difference between attenuation for apical vs basal sodium spikes with respect to branching order (*Figure 5—figure supplement 1F*; apical attenuation: 64.93±26.28 mV, basal attenuation: 49.72±24.3 mV, p-value 0.1997) or distance from the soma (*Figure 5—figure supplement 1E*; apical attenuation: 70.94±22.85 mV, basal attenuation: 49.72±24.3 mV, p-value 0.0844). Once again, however, we observe unusual attenuation values for basal dendritic segment #3 (all basal dendritic segments: 49.72±24.3 mV, outlier: –8.308 mV, basal dendritic segments excluding outlier: 59.4±5.8 mV), caused by additional current provided by its child branches that feature much lower sodium spiking thresholds (6 synapses for child branches vs 19 synapses for parent branch; see *Extended methods,* Relationship of branch order and distance with signal attenuation for details). As such, we repeated these two analyses excluding this data point. Results did not change for high apical dendrite branch order (apical attenuation: 64.93±26.28 mV, basal attenuation: 59.4±5.8 mV, p-value 0.2894), but a statistically significant difference did emerge in the comparison of distant apical dendrites against basal dendrites (apical attenuation: 70.94±22.85 mV, basal attenuation: 59.4±5.8 mV, p-value 0.0336).

The above suggest that signals originating in apical vs. basal dendritic segments with similar morphological characteristics (length, diameter, volume), attenuate more than those originating in basal dendritic segments. This is partly because most apical dendrites are located further away

from the soma compared to basal dendrites: comparison of EPSP and sodium spike attenuation with respect to distance from the soma yields statistically significant differences between basal and distant apical dendrites for EPSPs, although the difference is not significant for spikes (*Figure 5—figure supplement 1C, E*).

Finally, we also examined the relationship between dendritic sodium spiking thresholds and the morphological and electrophysiological parameters stated previously. We found that dendritic sodium spiking thresholds are significantly lower for apical versus basal dendritic segments with approximately equivalent values of electrotonic length (*Figure 5C*; p-value 0.001585), but this relationship is reversed for dendritic segments with similar volume (*Figure 5D*; p-value 0.000002). On the contrary, apical and basal dendritic segments featuring similar distances from the soma (data not shown; p-value 0.320436), dendritic path volume (data not shown; p-value 0.418108), length (data not shown; p-value 0.407686), or diameter (data not shown; p-value 0.592490) did not exhibit significant differences in their sodium spiking thresholds.

Taken together, this analysis shows that – as expected – the attenuation of signals originating in morphologically and electrophysiologically equivalent dendritic segments is significantly larger for apical vs. basal dendrites, likely because most apical dendrites are located further away from the soma. However, the threshold for inducing sodium spikes in apical dendritic segments is significantly smaller compared to basal dendritic segments of similar electrotonic length (most likely due to the proximity of the soma, causing a 'current sink' effect for basal dendrites), making it easier to drive such spikes in the apical compared to the basal tree. These findings provide a more intuitive explanation of the greater role of sodium conductances in the apical vs. basal trees of our model neuron.

## Discussion

In this work, we used a detailed computational model to delineate the dendritic constituents of neuronal computation in L2/3 V1 pyramidal cells. We found that the model exhibits a wide range of non-linear behavior at the dendritic segment level (*Figure 1*) and can mimic the activity patterns of neurons under spontaneous conditions in vivo (*Figure 2—figure supplement 1*). The model neuron exhibits robust orientation tuning in response to tuned input (*Figure 2A*), and under conditions of dendritic disparity, the model still remains robustly tuned (OSI <0.2 and tuning width >80°) for non-extreme dendritic disparity conditions (disparity ≤50°). Importantly, the neuronal orientation preference deviates from the expected one (*Figure 2D–F*), indicating that apical and basal inputs interact in non-linear ways. This bias is dependent on the distribution of synaptic inputs along the two trees (*Figure 2D–F*), it follows the preference of basal tree inputs under physiological conditions (*Figure 2D*) and is in line with prior work, whereby apical tree ablation had no impact on somatic orientation preference (*Figure 2—figure supplement 5*; *Park et al., 2019*). Remarkably, while the above would seem to suggest a minor role for apical inputs in the orientation preference of our model cell (*Figure 2D*), we found that the orientation tuning of the neuron is greatly affected by sodium conductances in the apical – but not the basal – tree (*Figure 4*). Interestingly, while basal sodium channel conductances are not critically important (*Figure 4C*), AMPA and NMDA conductances are necessary in both dendritic trees for orientation tuning (*Figure 4A, B*). This indicates that depolarizations from the basal tree play a key role in shaping neuronal output, even when basal sodium spikes do not. As such, neuronal output is a synergistic phenomenon mostly between apical sodium spikes and basal depolarizations or sodium spikes. Our simulations suggest that a form of *intra-tree dendritic cooperativity* – a synergistic effect between apical and basal dendritic segments that exhibit activity in relative synchrony – might be present in these neurons. This is supported by the existence of extinction-prone 'cooperative' somatic spikes that are lost when either of their dendritic components (apical or basal, be that depolarization or spiking) is lost (*Figure 3*). Additional evidence is provided by the experiments in which removal or suppression (decrease in conductance) of basal stimulus-driven synapses (i.e. only background-driven synapses are fully active) significantly decreases the quality of orientation tuning (i.e. OSI values) in the model neuron, or abolishes tuning altogether (*Figure 4A, B*).

A potential explanation for the differential contributions of apical and basal dendrites lies with their morphological and electrophysiological properties. Despite featuring a uniform sodium conductance throughout both dendritic trees, basal dendrites have higher sodium spiking thresholds than apical dendrites with similar electrotonic length values. Moreover, apical dendrites have highly variable spiking thresholds (*Figure 5D*). This is likely because the soma acts as a current sink, effectively

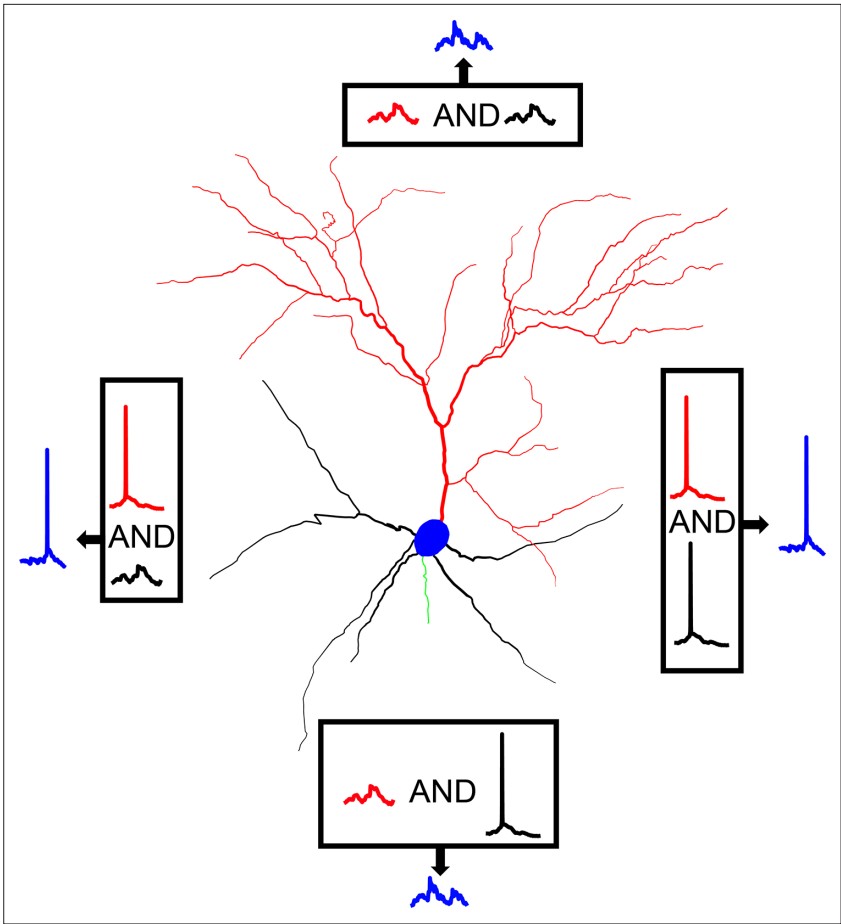

**Figure 6.** Graphical representation of AP generation via bimodal Input coincidence in a L2/3 V1 pyramidal neuron model. Action potential generation requires the spatiotemporal coincidence of apical sodium spikes with either basal sodium spikes or significant basal depolarizations, allowing the neuron to respond to salient stimuli that may or may not be affected by attentional and/or predictive signals from higher-order areas.

increasing the sodium spiking threshold of all soma-proximal branches (mostly basal dendrites). Meanwhile, the majority of apical dendrites feature much better electrical isolation from the soma due to distance and branching, and are as such exempt from such current sink effects, instead only suffering a loss of current from other connected apical dendritic branches. As such, they exhibit a much greater range of possible thresholds compared to basal dendrites of similar volume (*Figure 5D*), while simultaneously featuring greater signal attenuation (*Figure 5—figure supplement 1C, E*). Put together, these results indicate that apical dendrites overcome their signal attenuation deficit (compared to basal dendrites) by having a greater propensity for dendritic sodium spike generation for different amounts of input. This suggests that sodium spikes are more difficult to induce in basal compared to apical dendrites, offering a potential explanation as to why apical sodium conductances contribute more to somatic output.

Taken together, our model predicts that somatic output in L2/3 V1 pyramidal neurons is likely to be determined through *Bimodal Input Coincidence*, a subcategory of coincidence detection (*Figure 6*). Specifically, we propose that the basal tree receives visual feedforward input, representing the information therein as a series of hyperpolarizations, depolarizations, and occasional dendritic spikes. Thus, visual input creates a basal 'backdrop' of depolarizations that represents visual information. At the same time, predictive and attentional signals from higher-order cortical areas reach the apical tree of the neuron, causing the generation of dendritic spikes that propagate to the soma and are temporally summed with any concurrent basal depolarization. In the event that visual input is non-existent, or of

a non-preferred orientation, the depolarization is minimal to none. Thus, the apical dendritic spike will not be significantly augmented through summation and will fail to produce a somatic response in the majority of cases. If there is visual input, however, especially of an orientation matching the preferred orientation of the basal tree, the aforementioned 'backdrop' will include multiple subthreshold depolarizations, perhaps even dendritic spikes. The apical dendritic spikes are thus likely to be temporally summed with these depolarizations and generate a somatic spike. As such, even though most somatic spikes will be generated through an apical tree dendritic spike, the cases in which this is possible in the first place will be dictated by the backdrop of depolarizations (and/or spikes) provided by the basal tree.

The findings presented here propose a more central role for the apical tree than previously suggested. Specifically, in *Park et al., 2019*, we found that orientation selectivity was not abolished by in vivo laser ablation of the apical tree in L2/3 mouse V1 pyramidal neurons, remaining essentially unchanged following a recovery period of ~1 day. Importantly, the model pr'sented here reproduces the in vivo ablation findings of the aforementioned study (see Extended methods, Ablation protocol; *Figure 2—figure supplement 5*). What is seen as a contradiction is resolved by considering that in both *Park et al., 2019* and the simulations presented here, apical dendrite ablation is assumed to be accompanied by homeostatic alterations that counteract the increase in input resistance from the loss of the apical tree. These homeostatic mechanisms keep the firing rate of the neuron the same pre- and post-ablation, as seen experimentally (*Park et al., 2019*). Similarly, the neuron can potentially compensate for the loss of input from one of its dendritic trees via homeostatic mechanisms that amplify the effect of synaptic inputs on the other tree, an approach followed by *Mel et al., 1998* in their efforts to demonstrate that V1 L5 pyramidal neurons were capable of exhibiting orientation tuned responses through either apical or basal inputs. Thus, the contribution of the apical tree to somatic spiking in the intact neuron can be obviated by homeostatic mechanisms following apical tree ablation.

Our findings suggest that L2/3 V1 pyramidal cells may perform salient feature extraction from visual input compatible with predictive coding (*Rao and Ballard, 1999*). When top-down signals reflecting attentional or predictive processes reach the apical tree, they 'highlight' appropriate parts of the signal 'landscape' generated by the visual input reaching the basal trees through the generation of apical dendritic sodium spikes. This could result in a neuron that is activated only when specific features are present in its receptive field, facilitating the processing of salient stimuli. Such phenomena have been observed experimentally as a result of attention (*Simons and Chabris, 1999*).

However, this work also suffers from a number of limitations. Our findings are inherently bound to the particular characteristics of the simulated morphology. To ensure their generality, further exploration of different neuronal morphologies of L2/3 V1 neurons is required. However, given that our findings correlate with specific anatomical and electrophysiological features of the model neuron (*Figure 5*, *Figure 5—figure supplement 1*), it would be possible to infer whether the behavior we observe in our model could be present in other neuronal morphologies as well. In addition, while repeating our analysis using other morphologies is beyond the scope of this study (validating model function and performing the extensive parameter exploration reported here is highly resource-intensive), our code and all analysis scripts are made publicly available, allowing interested scientists to apply our approach to other morphologies. The sensitivity analysis that we added in Sensitivity analysis (*Figure 4—figure supplement 1*) also further establishes the robustness of our findings within biologically relevant ranges of ionic conductances. We have taken all these steps in order to more conclusively demonstrate that our findings are likely to generalize to other L2/3 V1 pyramidal neurons. Furthermore, our model is limited by the lack of data on dendritic features such as signal attenuation and distribution of ion channel conductances. However, this has been counteracted at least to some extent by the use of experimental data (e.g. see *Figure 2—figure supplements 1–4*) and the sensitivity analysis, which demonstrates that our conclusions are robust to biologically relevant variations of the key conductances (*Figure 4—figure supplement 1*). It should be noted, however, that there could be multiple 'solutions' to the problem of model validation, that is multiple different sets of values for the model parameters that yield the desired output. As such, we cannot be absolutely certain that our particular configuration of parameter values is correct, although this is an issue faced by all models in general, and not exclusive to this work.

Additionally, the single-neuron model sacrifices much of the complexity found in the actual L2/3 network, particularly certain input features. First, we do not simulate in vivo-like input patterns, due

to a lack of data. Instead, we use randomly generated Poisson spike trains, which are widely used in the field. Also, in our model, inhibitory inputs are not patterned according to different interneuron subtypes and are background-driven, lacking tuning and featuring firing rates lower than stimulus-driven excitatory synapses. A similar complexity loss is incurred by using a relatively simple stimulation protocol, compared to the alternating stimulation of the 'drifting gradient' protocol that is more widely used. Additionally, the model does not account for retinotopically shifted inhibitory inputs (e.g. *Rossi et al., 2020*). Retinotopic shift in inhibitory inputs would not affect our results, since stimulation is implicitly assumed to only take place at the center of the receptive field of the model neuron, a fact which would render any retinotopically offset neurons quiescent. Finally, although inhibitory inputs could be biased towards one dendritic tree over the other, the overall effect of such an imbalance would be a change in the net amount of excitation received by each tree – a scenario which we already explore in part (*Figure 2* and *Figure 3*), finding tuning and somatic spike generation to be generally unaffected. The omission of these features that were unlikely to influence the presence or absence of orientation tuning was performed in an attempt to reduce the complexity (and thus computational requirements) of the model.

In terms of questions that still need to be answered, the exact nature of the attentional and predictive signals received by the apical tree remains to be deciphered. Secondly, the formation of the visual 'backdrop' as a result of basal tree depolarization merits further study. Encoding of visual information using mostly subthreshold depolarizations in the presence of noise is an interesting problem, possibly resolved through the effects of intra-tree dendritic cooperativity, where apical spikes sharpen responses to stimulus-driven input rather than noise, improving the signal-to-noise ratio (*Poleg-Polsky, 2019*). In addition, a number of differences emerge when comparing the visual system across different model species. In rodents, orientation-selective cells akin to V1 simple cells have been discovered in the thalamus itself (*Scholl et al., 2013*), and direct thalamocortical projections to L1 of V1 have also been observed (*Roth et al., 2016*). Furthermore, rodent V1 organization follows a dispersed salt-and-pepper motif, unlike the highly structured organization typical of higher mammals like primates and cats (*Ohki and Reid, 2007*). These differences are important to take into account when trying to infer general rules of function from information derived from rules of information processing in different animal models. Finally, perhaps the most interesting unanswered question is whether these computations take place elsewhere in the cortex in addition to the visual system. Predictive coding as a means of stimulus compression has already been proposed as a way to simplify visual perception (*Rao and Ballard, 1999*), and this might also be the case for other sensory or cognitive tasks. Larger-scale models of the visual system, featuring realistic input structure and properties, will most likely be required to explore these phenomena in depth. Regardless, further investigation is required in order to unravel the Gordian knot that is visual perception.

## Acknowledgements

This work was supported by the ERC Starting Grant dEMORY, GA 311435, the Fondation Santé, a FORTH-Synergy Grant (EVO-NMDA), the H2020-FETOPEN-2018-2019-2020-01 NEUREKA, GA 863245 and the H2020 MSCA ITN 2019, SmartNets, GA 860949 to YP, the NIH NEI R01 grant EY-024019 and NINDS R21 grant NS088457 to SS, a Hellenic Foundation for Research and Innovation (HFRI) PhD Scholarship (no. 6731) to K-E.P, and the FORTH-Synergy Grant (FlexBe) to AP.

## Additional information

### Competing interests

Panayiota Poirazi: Reviewing editor, eLife. The other authors declare that no competing interests exist.

## Funding

| Funder | Grant reference number | Author |
|---|---|---|
| ERC Starting Grant dEMORY | GA 311435 | Panayiota Poirazi |
| H2020-MSCA ITN 2019, SmartNets | GA 860949 | Panayiota Poirazi |
| Fondation Santé a FORTH-Synergy Grant | EVO-NMDA | Panayiota Poirazi |
| H2020-FETOPEN-2018-2019-2020-01NEUREKA | GA 863245 | Panayiota Poirazi |
| NIH NEI | R01 grant EY-024019 | Stelios Smirnakis |
| NINDS | R21 grant NS088457 | Stelios Smirnakis |
| Hellenic Foundation for Research and Innovation | PhD Scholarship (no. 6731) | Konstantinos-Evangelos Petousakis |
| FORTH-Synergy Grant | FlexBe | Athanasia Papoutsi |

The funders had no role in study design, data collection and interpretation, or the decision to submit the work for publication.

## Author contributions

Konstantinos-Evangelos Petousakis, Data curation, Software, Formal analysis, Validation, Visualization, Methodology, Writing – original draft; Jiyoung Park, Resources, Data curation, Writing – review and editing; Athanasia Papoutsi, Conceptualization, Supervision, Writing – original draft, Writing – review and editing; Stelios Smirnakis, Conceptualization, Resources, Supervision, Writing – review and editing; Panayiota Poirazi, Conceptualization, Supervision, Funding acquisition, Writing – original draft, Project administration, Writing – review and editing

## Author ORCIDs

Konstantinos-Evangelos Petousakis ⓘ http://orcid.org/0000-0003-2022-1671
Athanasia Papoutsi ⓘ http://orcid.org/0000-0003-2466-7259
Stelios Smirnakis ⓘ https://orcid.org/0000-0002-1929-2811
Panayiota Poirazi ⓘ https://orcid.org/0000-0001-6152-595X

## Decision letter and Author response

Decision letter https://doi.org/10.7554/eLife.91627.sa1
Author response https://doi.org/10.7554/eLife.91627.sa2

# Additional files

## Supplementary files

• MDAR checklist

## Data availability

The code used to create the figures of this manuscript is publicly available in GitHub: https://github.com/kepetousakis/petousakis_etal_2023 (copy archived at *Petousakis, 2023*). The code for the model neuron is available in ModelDB at https://modeldb.science/267501.

The following dataset was generated:

| Author(s) | Year | Dataset title | Dataset URL | Database and Identifier |
|---|---|---|---|---|
| Petousakis KE, Park J, Papoutsi A, Smirnakis S, Poirazi P | 2023 | L2/3 V1 Pyramidal Cell model (modified Park et al., 2019; a/n: 231185) (Petousakis et al., 2023) | https://modeldb.science/267501 | ModelDB, 267501 |

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

# Appendix 1

## Extended methods

### Model description

The single neuron model was derived from the L2/3 V1 pyramidal cell model of created in the NEURON simulation environment (*Hines and Carnevale, 2001*). It is a morphologically detailed reconstruction, featuring 43 apical dendritic segments, 7 basal dendritic segments and an axon. As the model makes use of a variety of passive and active mechanisms (*Tables 1–4*), its electrophysiological properties were validated against experimental data (*Park et al., 2019*). Neither the morphology nor the biophysical properties of the source model were altered, barring specific ion channel or synaptic changes where explicitly specified.

The model features both excitatory and inhibitory synaptic input. Excitatory input was subdivided into background-driven (noise) and stimulus-driven, the latter consisting of orientation-tuned synapses. Inhibitory synapses are modeled as background-driven (inhibitory noise) which alongside background-driven excitatory synapses (excitatory noise) comprise the background-driven inputs. Stimulus-driven and background-driven synaptic input to the neuron is distributed uniformly on the target compartments using a density of 2 synapses per μm (*Defelipe and Fariñas, 1992*; *Schuz and Palm, 1989*). Activation of synapses occurs via pseudo-randomly generated Poisson spike trains. Background-driven and stimulus-driven synapses feature different peak Poisson activation frequencies (0.11 Hz and 0.3 Hz, respectively) to produce the electrophysiologically recorded firing rates (*Adesnik et al., 2012*; *Park et al., 2019*). Stimulus-driven synapses feature variable activation frequencies, with a maximum of 0.3 Hz, based on the angular distance of the stimulus from the preferred orientation of each individual synapse. Unlike the original model by *Park et al., 2019*, apical stimulus-driven synaptic inputs have had a 10ms delay added, to better reflect experimental data on feedback synaptic inputs in V1 (*Ju et al., 2020*).

We modulate the proportion of stimulus-driven excitatory synapses on the apical versus the basal tree in various experiments in order to examine the effects of synaptic input distribution on somatic output. In all cases, the amount of inhibition at the soma is 7% of the total inhibitory synapses (*Defelipe et al., 2002*). The default synapse distribution was 40% apical to 60% basal ("biologically plausible model"), in adherence with experimental data (excitation: 40% apical, 60% basal; inhibition: 33% apical, 60% basal, 7% soma) (*Defelipe et al., 2002*). The 'inverse biologically plausible' model, conversely, features a 60% apical to 40% basal distribution (excitation: 60% apical, 40% basal; inhibition: 55.8% apical, 37.2% basal, 7% soma). The 'even' model features a balanced distribution of synaptic input across both trees (even: excitation: 50% apical, 50% basal; inhibition: 46.5% apical, 46.5% basal, 7% soma). The synaptic distribution of the 'even' and 'inverse biologically relevant' models are not derived by experimental data, yet they allow for the investigation of apical and basal tree interactions under conditions of balanced or imbalanced synaptic input. We also make use of other distributions biased in either the apical or basal direction (referring to stimulus-driven excitatory input; from 10% basal – 90% apical to 90% basal – 10% apical, in steps of 10%). Similar to above, background-driven synapses closely follow the stimulus-driven synapse distribution in these cases, except for background inhibition, which always features 7% allocated to the soma.

In our model, orientation preference was assigned on two levels: synaptic (input/single synapse) and dendritic (whole tree). Single synapse orientation preference is modeled as a synaptic weight vector biased towards a predefined (preferred) orientation, which also factors into the activation frequency of orientation-tuned synapses (*Chen et al., 2013*). Each synapse is assigned an orientation preference, with values being 0° to 360° in steps of 10°. This distribution of orientation preferences to the stimulus-driven synapses at the dendritic tree level followed a Gaussian probability density function wrapped around the unit circle:

$$f(s) = \frac{1}{W\sqrt{2\pi}} \sum_{k=-1}^{1} e^{\frac{-(s-b+2\pi k)^2}{2W^2}}$$

where W is the standard deviation (30°; *Chen et al., 2013*), s is the synaptic orientation preference and b is the mean of the distribution (apical or basal tree mean orientation preference, ranging from 0° to 90° in steps of 10°). The parameter k controls the wrapping around the unit circle, and takes

the values [-1,–0.5, 0, 0.5, 1]. As a result, the mean orientation preference of each dendritic tree is biased towards the chosen mean orientation preference. By implementing excitatory stimulus-driven connectivity in this way, we also emulate the 'like-to-like' connectivity motifs seen in vivo (as seen in e.g. *Ko et al., 2011*; *Rossi et al., 2020*), as (virtual) presynaptic neurons with orientation preferences close to the mean orientation preference of the model neuron are more likely to connect to it, due to being over-represented in the overall pool of possible synapses. Somatic/neuronal orientation preference was evaluated using the approach described in section 2.3.3.

### Biophysical manipulations

The following manipulations aimed to evaluate somatic output under different sets of conditions:

1. Sodium (channel) blockage: selectively nullify (set to zero) sodium conductance ($g_{Na}$) in a designated compartment (e.g. apical tree, specific basal dendritic segments, or soma).
2. Synaptic modulation: selectively alter synaptic conductances ($g_{AMPA}$/$g_{NMDA}$) in a designated compartment or dendritic tree.
3. Input manipulation: selectively deactivate specific types of input (excitatory stimulus-driven and background-driven).

### Simulation protocols

All simulations were performed on our High-Performance Computing Cluster (Rocks 7.0) with 624 cores and 3.328 TB of shared RAM under CentOS 7.4 OS (*Papadopoulos et al., 2003*), through the NEURON simulation environment (*Hines and Carnevale, 2001*). Model neuron output measurements were obtained at a sampling rate of 10 KHz (dt = 0.1ms).

The following simulation protocols were implemented:

### Iterative paired-pulse protocol (I3P)

Used for the evaluation of dendritic non-linearities. Following synaptic silencing and $g_{Na}$ nullification in the entire model neuron excluding the dendritic segment under investigation, a number (1–200, with step 1) of evenly spaced (uniformly distributed) excitatory synapses are synchronously activated (2 pulses at 50 Hz). Voltage at the midpoint of each dendritic segment was recorded to evaluate dendritic non-linear behavior caused by sodium and NMDA spikes.

### Stimulation protocol

The model is simulated for 2500ms (2.5 s), with oriented stimulus presentation onset at 500ms (0.5 s), lasting until the end of the simulation (duration 2 s).

### Orientation tuning validation protocol

Using the above stimulation protocol, we simulate 10 model 'neurons' that feature different synaptic location patterns, defining 10 distinct *neuron IDs*. For each one of these neuron IDs, we use 10 different permutations, with different instances of the input spike trains, resulting in 10 different *simulation IDs* for each neuron ID. Stimuli of 18 different orientations (0° to 170° in steps of 10°) are then presented in separate simulations. The tuning curve is extrapolated by averaging firing rates across simulation IDs for each neuron ID. The overall tuning properties (width, OSI) were then computed through averaging responses across neuron IDs for each stimulus orientation. Neuron IDs with an OSI of less than 0.2 or a tuning width greater than 80° (as determined by a Gaussian fit) are excluded. If more than 20% of neuron IDs do not meet this criterion, the specific parameter combination is considered to not result in orientation tuning. No neurons were excluded based on the preceding criteria in our simulations.

### Orientation disparity protocol

To evaluate the contribution of the apical and basal trees in the somatic orientation preference, we keep the mean apical tree orientation preference fixed at 0°, and set the mean basal tree orientation preference to any one of 10 different values (0° to 90° in steps of 10°) (see also Sensitivity analysis). Thus, we introduce orientation tuning *disparity* between the two trees, ranging from 0° to 90°. Akin to the orientation tuning protocol, we independently generate 10 different neuron IDs with 10 different simulation IDs for each one and present stimuli as above (Orientation tuning validation protocol). Orientation preference is then calculated as described above (Orientation tuning validation

protocol). If more than 20% of the neuron IDs are rejected (OSI <0.2 or tuning width >80°) for a given degree of orientation tuning disparity, then the model as a whole is deemed to exhibit no orientation tuning for that level of disparity.

In order to determine whether the somatic response follows the mean apical or basal tree preferred orientation most closely, we compute the *expected* orientation preference of each neuron ID; that is, the orientation preference assuming a linear summation of the synaptic inputs. Specifically, we record the assigned orientation preference of all synapses per neuron ID, and then we use Euler's Formula to calculate the expected orientation preference for this neuron ID:

$$E_d = arctan$$

Where $E_d$ is the expected orientation preference for a degree of disparity d, i and j are the numbers of stimulus-driven apical and basal synapses respectively, and $T^A$ and $T^B$ are lists of synaptic orientation preferences for the apical and basal trees, respectively. In this particular protocol, a remapping of preferences is required in specific cases (e.g. 180° → 0°) since we are not simulating direction selectivity, projecting orientation preferences into the [–90°, 90°] interval.

## Spontaneous activity protocol

Similarly to the orientation tuning validation protocol, we use 50 different neuron IDs with 100 different simulation IDs for each one. Simulations feature no stimulus-driven synaptic activity, leaving only background-driven synapses active.

## Ionic intervention protocol

Using 50 neuron IDs with 10 different simulation IDs for each one, prior knowledge of action potential occurrence is obtained via the stimulation protocol (see Stimulation protocol; presented stimulus: 0°, corresponding also to the mean apical and basal tree orientation preference), using the 'biological plausible' synaptic distribution (Model description). This protocol proceeds similarly to the stimulation protocol (with the notable exception of somatic $g_{Na}$ being set to 0 to prevent backpropagation) until a time before point $t_i$, which is the time of somatic spike occurrence. At $t_i$ – 3ms, $g_{Na}$ is set to 0 on either the apical or basal tree dendrites in two separate simulations, until $t_i$ +1ms. This manipulation selectively eliminates the contribution of apical or basal dendritic sodium conductances (seen either as depolarization or as dendritic spikes) to the somatic spike. This protocol is repeated for all somatic spikes and the ensuing voltage traces are analyzed to ascertain whether the somatic spikes survived the manipulation. Spikes rendered extinct only when intervening on the apical tree are labeled as 'apically driven', while those rendered extinct when intervening on the basal tree are labeled as 'basally driven'. The spikes rendered extinct by either intervention are labeled as 'cooperative'. We observed no cases of somatic spikes failing to become extinct in at least one of the two interventions while using this protocol.

## Synaptic intervention protocol

This protocol proceeds similarly to the ionic intervention protocol (note that somatic $g_{Na}$ is not set to 0 in this protocol), with two main differences: (a) the intervention reduces stim-driven AMPA and NMDA synapse conductances on either the apical or basal tree by a specific percentage (e.g. 50%, 90% etc.), rather than setting $g_{Na}$ to 0, (b) the window of the intervention changes from [$t_i$ – 3ms until $t_i$ +1ms] to [$t_i$ – 30ms until $t_i$ +10ms] (10 x increase in window size). This was done to compensate for the relatively low peak activation frequencies of stim-driven synapses (0.3 Hz).

## Ablation protocol

Apical dendrites and their corresponding synapses are not generated under this protocol, simulating a neuron 1 day after laser ablation of the apical dendrites (*Park et al., 2019*). As firing rates greatly increase (more than 10-fold increase) under these conditions, basal AMPA and NMDA synaptic conductances are reduced by 45% to bring the firing rate back down to expected levels.

## Two-photon microscopy

### Animals in experiments

All experimental protocols were approved by Brigham and Women's Hospital (BWH) Institution Animal Care and Use Committee. Male and female wild-type (C57BL/6) mice were purchased from

The Jackson Laboratory (RRID:MGI:2159769) and bred for experiments. A total of two mice were used in these experiments.

## Chronic window implantation and sparse labeling

Viral vectors were purchased from Addgene. For viral injections and chronic window implantation, 6- to 10-week-old wild type C57BL/6 mice (both male and female) were anesthetized with isoflurane (1–1.5%). The depth of anesthesia was assessed via monitoring breathing rates. Meloxicam (2.5 mg/kg) and Buprenorphine (0.1 mg/kg) were administered subcutaneously every 12 hr following the surgical procedure for 2 days. A headpost was implanted on the skull and a 3 mm diameter craniotomy was made over the visual cortex of the left hemisphere. The craniotomy was centered 2.7 mm lateral to the midline and 1.5 mm posterior to the bregma. To sparsely label pyramidal neurons with GCaMP6s, a mix (~90 nl per site, up to 2 sites) of diluted CamKII-CRE (AAV1, diluted 80,000 X) and flexGCaMP6s (AAV5, diluted up to 2 X) was injected slowly over 5 min per penetration using a Drummond Nanoject III. Up to two penetrations ~0.5 mm apart on average were performed per craniotomy. After the viral injection, a round coverslip was fitted to the craniotomy and sealed with vetbond and dental cement. Most chronic windows in our hands remain clear for 2–3 months. Two-photon imaging was performed on weeks 3–4 following viral injection, at which time GCaMP6s expression was optimal.

## In vivo calcium imaging of sparsely labeled neurons

Three to four weeks after the injection, each GCaMP6s expressing mouse was sedated with Fentanyl (0.5 mg kg$^{-1}$) and Dexmedetomidine (0.5 mg kg$^{-1}$) (*Mrsic-Flogel et al., 2007*) for imaging experiments. A stable level of anesthesia was confirmed by stable breathing rate and lack of movement. Neuronal and dendritic calcium activity was recorded from L2/3 pyramidal neurons in V1. We chose neurons with 1–3 primary basal dendritic segments clearly visible up to their primary branching point on the same plane with the soma. Apical dendrites were not present in the focal plane during recording. Mice were in the dark during the imaging process and their spontaneous firing rate was recorded. Images were acquired at ~7.3 frames/s using an Ultima IV microscope in spiral scanning mode with a 25X1.0 NA Nikon objective (5–30 mW laser power, 900 nm). Recordings lasted for 283 s (2750 frames). The size of the recording FoV was 103 μm$^2$, and the pixel size was 0.201 μm$^2$. Twelve pyramidal neurons were imaged, and 11 of these were selected for analysis. The final neuron was not selected due to an insufficient number of basal dendritic segments being positively identified as belonging to said neuron following visual inspection of a z-stack image.

## Data analysis

Data analysis of both modeling and two-photon imaging experiments was performed through Python 3.8+, using publicly available libraries as well as custom-made scripts and functions. The latter include:

## Dendritic spike detection

Dendritic sodium spikes were identified via a detection algorithm that identified depolarizations exceeding a –20 mV threshold. For the identification of NMDA spikes in the I3P experiments (see Iterative paired-pulse protocol (I3P)) we located the inflection points (points where the second derivative is zero) of the voltage trace after the second pulse. As the NMDA spike exhibits a voltage plateau, 2 or more inflection points indicate an NMDA spike. As our voltage measurements are not continuous and the I3P protocol does not feature noise, it is unlikely to encounter a measurement point at which the second derivative is exactly zero. Thus, we assume an inflection point exists at some recording point $P_n$ if and only if $(P_{n-1} \cdot P_{n+1}) < 0$, where $P_{n-1}$ and $P_{n+1}$ refer to existing points in the second derivative of the voltage trace, immediately preceding and following the theorized inflection point.

## Quantification of dendritic nonlinearities

Using data from the I3P experiments, we calculated the maximum amplitude of dendritic segment depolarization per synapse count. A linear input-output curve is then generated, extrapolating from the response of the dendritic segment to a single synaptic input (unitary response). This curve is compared to the actual response of the dendritic segment to increasing input in an 'Expected vs. Actual' plot, thus coarsely characterizing the integration mode of the dendritic segment. The integration mode of the dendritic segments was quantified in detail using the Nonlinearity Relative

to Linear Extrapolation (NRLE) metric (*Behabadi et al., 2012*), which is defined as the maximum ratio of actual to expected neuronal output signal. An NRLE value of less than 1 denotes a sub-linear dendritic segment, NRLE of exactly 1 indicates a linear dendritic segment and NRLE values over 1 characterize a dendritic segment as supra-linear. The non-linearity threshold of all dendritic segments was also calculated, measured as the minimum number of synapses required to elicit the corresponding electrogenic event (sodium or NMDA dendritic spike).

## Two-photon imaging data analysis

Recordings are motion corrected via the suite2p software (*Pachitariu et al., 2016*) to accurately extract fluorescence values from dendritic segments. In the case of somatic activity data, we use the original video, as motion correction is not necessary at the level of magnification used. Using the Fiji ImageJ software (*Schindelin et al., 2012*), ROIs are manually defined, using a maximum intensity z-stack as a reference image. Dendritic segments are identified as belonging to the cell of interest via visual inspection of a z-stack recording. Each dendritic segment is broken up into multiple ROIs. Only one ROI is dtfined for the soma. We completely avoid areas where there is an overlap of the cell of interest with nearby dendrites/axons of cells from outside the field of view. Dendritic ROIs are slightly wider than the dendritic segments themselves, to partially account for misplacement of the ROI or uncorrected movement (drift) of the dendrite. 4 equal-sized circular ROIs are also defined in areas where no visible dendritic segments or somata can be found to calculate the background fluorescence level.

## Calcium trace processing

Raw calcium fluorescence data is normalized via a moving window average of the fluorescence trace (window size of 20 s), subtracting $F_0$ from it (defined as the average of the 20% lowest fluorescence values of said moving 20 s window), then dividing by the same value (F0), finally producing a normalized $\Delta F/F_0$ value. These $\Delta F/F_0$ values were significantly different from the background $\Delta F/F_0$ as calculated from the 4 circular reference ROIs (cross-correlation test; p<0.05; no manually defined ROIs were excluded through this process).

## Calcium event detection

Calcium events are detected via a simple dual-threshold approach: a calcium event is registered when the $\Delta F/F$ value exceeds the 93.5th percentile at the same time as the derivative of the $\Delta F/F$ value exceeds the 67th percentile. These percentile values were calculated via trial-and-error, attempting to match the expected firing frequency of a neuron (soma) under spontaneous activity conditions (approximately 0.2 Hz; *Haider et al., 2013*). To ensure that the algorithm returns true events 'ather than random noise, we performed an Event-Triggered Average (ETA) analysis. We extracted all of the detected event waveforms, centered them on the time of detection, and averaged across waveforms. We did this separately for somatic and dendritic events (*Figure 2—figure supplement 1C*), as well as for each animal separately (*Figure 2—figure supplement 3A, B*). Calcium event-triggered averaging of the processed fluorescence signal yields the expected temporal profile, indicating that these events are not likely to represent noise.

## Generation of 'fluorescence-like' traces from model voltage traces

Fifty neuron IDs were simulated with 100 simulation IDs each, using the 'biologically plausible' configuration under the spontaneous activity protocol, in order to closely match the properties of the real neurons under spontaneous activity conditions. Next, we ran a spike detection algorithm to find all somatic and dendritic spiking events. We binarized the traces, using 1 for spike peaks and 0 otherwise. We then convolved these binarized traces with a GcaMP6s kernel function of our own design:

$$F\left(t\right) = -s\left(e^{\frac{-t-t_0}{t_r}} - e^{\frac{-t-t_0}{t_d}}\right)$$

where t is time (in ms), s is a scaling factor (s ≈ 0.5714), t0 is the time of the event, $t_r$ is the rise time ($t_r$ = 74ms) and $t_d$ is the decay time ($t_d$ ≈ 401.45ms). The output of the function is unitless, representing fluorescence. With these parameters, our kernel function has an actual rise time constant of 194.225ms, with output values decaying by ~52.11% within 0.6 s (*Figure 2—figure supplement*

**2A**). These values closely match experimentally observed values for GCaMP6s (**Figure 2—figure supplement 2B**; **Chen et al., 2013**). The ensuing fluorescence traces were then downsampled to a dt of 0.138 s (~7.25 Hz) from the original dt of 0.1ms (10 KHz), to emulate the sampling rate of our two-photon calcium fluorescence imaging. The transformed traces were then processed by the same pipeline as true calcium fluorescence data (see Calcium trace processing and Calcium event detection).

## Analysis and comparison of true vs 'generated' fluorescence data

Voltage traces from the soma and basal dendritic segments were converted into approximations of two-photon fluorescence traces (see Generation of "fluorescence-like" traces from model voltage traces) to allow a direct comparison with in vivo calcium imaging data that were collected under similar conditions. Significant fluorescence increase 'events' were identified for both datasets using a custom detection algorithm (see Calcium event detection). To ensure that the algorithm detects true events rather than noise in the two-photon signal, we performed an event-triggered averaging (ETA) analysis of trace segments centered on the event onset timepoint (see Calcium event detection; 1131 dendritic events, 96 somatic events across all 11 neurons and their respective dendritic ROIs). The results of this analysis indicate that random noise is unlikely to be the cause of the detected events, as there is a marked increase in fluorescence intensity at the time of event onset for all detected events (**Figure 2—figure supplement 1D**, **Figure 2—figure supplements 3–4**) which persists over time.

Next, in order to infer the relationship between basal and somatic events, we looked at their relative timing differences. Using dendritic events as the starting point, we paired each dendritic event with the most temporally proximal somatic event, and computed the time difference between the two for each event pair (measured in imaging frames, each lasting approximately 138ms). Using the same approach, we measured the fraction of somatic-dendritic event pairs in our model and compared them to the in vivo data. Event pairs featuring a temporal distance of zero (occurring within the same frame) were excluded as they are most likely to represent backpropagating events (excluded event pairs: model: 219,291/245,717 event pairs, 89.25%; in vivo: 29,143/237,572 event pairs, 12.27%; difference attributed to a lack of noise-related fluctuations in the model data compared with the in vivo data). The resulting distributions (**Figure 2—figure supplement 1E, F**) show that in both the model and in vivo data local dendritic events can be generated independently of somatic events. Paired events separated by more than 4 recording frames (~552ms) are highly unlikely to be the result of backpropagation, even when considering the slow kinetics of the calcium sensor. Moreover, the somatic events that are preceded by basal dendritic events are similar in proportion between datasets (**Figure 2—figure supplement 1E, F**, red bins: 48.79% model, 48.12% actual when including all event pairs with a non-zero timing difference; 24.3% model, 27.8% actual when including only event pairs with an absolute timing difference >4 recording frames).

To mitigate the risk that the in vivo findings are the result of noise or random chance, we constructed a null distribution of timing differences for the in vivo data by independently circularly shuffling the time-series data from the somatic and various dendritic segment ROIs 1000 times, and then re-computed the timing differences. We then compared the actual data with the null distribution data using the two-sample Kolmogorov-Smirnov (KS) test, finding that the two distributions of timing differences were significantly different (**Figure 2—figure supplement 4C**; test statistic ~0.105, $p \ll 0.05$), whereas a similar analysis of completely randomly generated event timings would results in identical (not statistically different) distributions for event pair timing differences. It is important to note that the analyses herein are inherently subject to the noisy nature of the recordings, the limitations of the slow kinetics of the calcium indicator, the low temporal resolution of the recording (frame time of ~138ms), as well as the possibility of undetected events.

## Electrotonic length calculation

The electrotonic lengths of all dendritic segments were calculated using the formula from **van Elburg and van Ooyen, 2010**:

$$\lambda_i = \sqrt{\frac{b_i r_m}{2 r_a}}$$

where i refers to the dendritic segment, $\lambda_i$ is the electrotonic constant, $b_i$ is the radius of the branch, $r_m$ is the specific membrane resistance and $r_a$ is the intracellular resistivity. To derive the final value, we divide the length of each dendritic section by its corresponding $\lambda_i$, which produces a dimensionless quantity.

## Quantification of dendritic impact on somatic output

To ensure relative equivalence between the levels of activity in each dendritic branch, we start by providing each dendritic branch the minimum amount of synaptic input required such that each one features either an EPSP of approximately 20 mV (19.78±0.98 mV; *Figure 5A, B*, *Figure 5—figure supplement 1C, D*), or a dendritic sodium spike (*Figure 5—figure supplement 1A, B*, *Figure 5—figure supplement 1E, F*). Only one dendritic branch received inputs each time, and somatic (but not dendritic) sodium channels had their conductance set to zero in order to accurately measure signal attenuation. After measuring the attenuation of this EPSP or spike at the soma, we attempted to describe it as a function of different morphological and electrophysiological parameters by dividing the attenuation observed at the soma by the corresponding value of the parameter for the given dendritic branch. The parameters used were dendritic length, dendritic diameter, distance of dendritic segment from the soma, dendritic volume, dendritic path volume (defined as the sum of the volumes of all dendritic segments linking the given dendrite with the soma, including the volume of the given dendrite) and dendritic electrotonic length (see Electrotonic length calculation). Values can then be compared across dendrite types (apical vs basal) for each parameter by performing Welch's t-test. We also performed a similar analysis using the dendritic sodium spiking threshold (*Figure 1E, F*) instead of dendritic signal attenuation, with the same morphological and electrophysiological parameters as above (*Figure 5C, D*). To ensure fairness of comparisons, we excluded from these analyses any dendrites belonging to the apical trunk (4/43 apical dendritic segments), any apical dendrites with atypical properties (4/43 apical dendritic segments featuring a relatively high sodium spiking threshold for their morphology, produced as a side-effect of the 3d reconstruction; lengths: 16.2±10.5 μm; diameters: 0.711±0.027 μm), as well as any apical dendrites whose value for the parameter under examination was outside 3 standard deviations of the basal dendrite mean for the same parameter. Note that the qualitative results of the analysis did not change even if no dendrites were excluded.

## Relationship of branch order and distance with signal attenuation

To check whether apical and basal dendrites feature different levels of attenuation for 20 mV EPSPs or dendritic sodium spikes (see Quantification of dendritic impact on somatic output), we compared the levels of attenuation for these events between basal dendrites and apical dendrites of high branch order (>3) or high distance from the soma (>182.7 μm, equivalent to the maximum distance of any basal dendritic segment from the soma). We excluded basal dendritic segment #3 (root branch of the bifurcating basal dendrite) from some analyses, since it yielded extremely low or even negative attenuation values because of additional current received from its much more excitable child branches (sodium spiking thresholds: 6 for child branches vs 19 for parent branch, basal dendritic segment #3; EPSP attenuation values: all basal dendritic segments: 12.69±4.04 mV, outlier: 5.11 mV, basal dendritic segments excluding outlier: 13.96±2.8 mV; spike attenuation values: all basal dendritic segments: 49.72±24.3 mV, outlier: –8.308 mV, basal dendritic segments excluding outlier: 59.4±5.8 mV). After isolating the EPSP and spike attenuation data from the dendrites that matched the above criteria, we compared the resulting sets of values using Welch's t-test.

