## [Editor Report]

This manuscript will be valuable to scientists working on visual neurobiology and cortical processing. It uses a compartmental model to evaluate the relative contribution of basal and apical dendritic trees to the orientation selectivity of layer 2/3 pyramidal cells. There is solid support for the key claims that pertain to the model itself, but there are some questions as to how well the model reflects the biological circuit.

---

## [Decision Letter]

**Decision letter after peer review:**

[Editors’ note: the authors submitted for reconsideration following the decision after peer review. What follows is the decision letter after the first round of review.]

Thank you for submitting the paper "A Tale of Two Trees: Modeling Apical and Basal Tree Contribution to L2/3 V1 Pyramidal Cell Orientation Selectivity" for consideration by *eLife*. Your article has been reviewed by 2 peer reviewers, and the evaluation has been overseen by a Reviewing Editor and a Senior Editor. The reviewers have opted to remain anonymous.

Comments to the Authors:

We are sorry to say that, after consultation with the reviewers, we have decided that this work will not be considered further for publication by *eLife*.

*Reviewer #1 (Recommendations for the authors):*

This work aims to determine the contribution of apical and basal dendrites to orientation selectivity in a cortical neuron. This is a relevant question because basal and apical dendrites receive different qualitatively different classes of inputs: feedforward inputs favour basal dendrites whereas feedback inputs preferentially target apical dendrites. Understanding the rules by which these two dendritic fields interact at the level of single neurons is therefore useful for understanding and predicting how different information streams might be integrated by cortical neurons. Here, the authors take a compartmental modelling approach. Using a previously established model of a layer 2/3 pyramidal cell, they systematically vary some of the model parameters to explore and quantify how much apical and basal dendrites contribute to the generation of somatic spikes when synaptic input is tuned to orientation. The main conclusion is that both apical and basal contribute to spiking, with the apical contribution depending more on sodium conductances and the basal contribution depending mostly on AMPA and NMDA receptors. This conclusion is well supported by the data and the parameter exploration presented will be useful to the field. While the data come from a compartmental model, attempts to understand the biophysical basis of the phenomena described are limited and often not grounded in the current knowledge of cortical neuron biophysics. This aspect of the manuscript could be significantly expanded.

Strengths:

The main strength of this paper is the systematic variation of parameters such as synapse distribution and background activity, and the subsequent quantification of the effect of changing these parameters on somatic spiking. The disparity experiments are interesting, especially for seeing when orientation selectivity breaks down, and extend the modelling results presented in Park et al. 2019. The temporally precise manipulations of conductances and direct spike-by-spike comparison between control and manipulation conditions is a new and nice approach that should be useful for future studies. Another potential strength is the attempt to match the background activity observed in vivo, though this effort has significant weaknesses.

Weaknesses:

Most of the results in the paper describe the behavior of the model when a set of parameters are changed. While this is useful information, the major strength of a model is that it should be possible to understand exactly why model behaves the way it does, but the authors do not make a serious attempt at this. Why do apical dendrites rely more on sodium conductances to influence somatic spiking? Why do basals rely more on AMPA and NMDA conductances? What exactly is different in terms of the spatiotemporal patterns of inputs and voltage propagation across the neuron during "apical", "basal" and "cooperative" somatic spikes? The risk here is that the model behavior might only be valid for a restricted set of parameters (eg: density and distribution of conductances, most of which are not known experimentally), and thus the generality of the findings might be limited. Understanding the biophysics should allow for deriving general rules for the problem studied here and more robust conclusions.

Figure 6 is a step in this direction, but it mostly shows that the number of synapses to reach threshold increases with cable diameter and decreases with distance from soma, which is well known (ie: where the passive and active properties of the cable are uniform, as they are here, the result is explained by the differences in input impedance). The lack of granularity is well illustrated in the title of section 3.6.: "Morphological and Electrophysiological Properties Influence Dendritic Behavior" – indeed they do, this is very well known. But how exactly in this model? Overall, I would have expected a more thorough analysis of the relationships between voltage propagation, morphology, and synaptic activation pattern. For example, could the main results here be explained by fast sodium spikes being more heavily filtered by the basal dendrites (which tend to have a shorter spatial constant, eg : Nevian et al. 2007), which would also explain the higher dependency of the basals on NMDA conductances, which are slower and therefore less filtered? Or are there differences in the relationship between synaptic input patterns and the generation of local non-linearities between basals and apicals?

The attempt to use experimental calcium imaging data to estimate background activity is nice, but the data are not very convincing. In Figure 2C, most of the isolated dendritic events look like noise. I realise that the event-triggered average shown in 3D suggests that there is some real signal being analysed, but I suspect that it is heavily contaminated by noise, and that the authors are grossly over-estimating the frequency of decoupled dendritic events. The method details of the analysis procedure are also worrying: "These percentile values were calculated via trial-and-error, attempting to match the expected firing frequency of a neuron (soma) under spontaneous activity conditions". What is the rationale here? The goal should be to detect the real events in the data, not changing the detection parameters until the result matches a previous publication. Also, if in the end the goal was to obtain the same value as other studies, it is unclear why the experiment was needed in the first place.

1. The authors propose that in Figure 1 they are validating their modelling approach, but it is unclear what exactly is being validated here. The figure shows that dendrites with sodium channels and AMPA/NMDA synapses can generate non-linear input-output curves, which is very much a guaranteed finding – in a model with reasonable passive parameters and morphology (which is the case here since model that has been previously validated against experimental data), I don't see how the result could be any different. Please clarify.

2. Line 587: figure call is wrong, should be 4A (same for the subsequent call)

3. In Figure 6 it is not clear why the results in panel C are qualitatively different that in A. Rm and Ra are presumably uniform, so the electrotonic constant is a transform of the dendritic diameter. Please explain.

*Reviewer #2 (Recommendations for the authors):*

The manuscript by Petousakis et al. describes a modeling study of a pyramidal neuron in the Layer 2/3 (L2/3) of mouse primary visual cortex V1. The authors use a biophysically detailed, spatially extended model of the neuron, with voltage-gated conductances in the soma, basal dendrites, and apical dendrites to investigate how inputs to basal and apical dendrites sculpt the activity of the neuron and the corresponding major computation that L2/3 excitatory cells presumably carry out – that of orientation detection. The authors report a number of computational experiments and conclude that inputs to basal and apical dendrites synergistically sculpt the orientation selectivity of the neuron.

While the premise is interesting, the study, and especially the model architecture, relied on a number of assumptions that limit the interpretability of the outcomes.

1. The whole study is based on a single model of one neuron. This is a model from Park et al., Nature Communications, 2019, which in turn is based upon other previously published models (e.g., the supplementary tables 2-4 in this manuscript are copied from the Park et al. paper). Park et al. already showed how one can get orientation selectivity with this neuron model and how it can be manipulated by ablating apical and basal dendrites.

2. There is a lot of data on function-dependent connectivity for L2/3 excitatory neurons that sculpts orientation selectivity, but the authors do not mention even the most prominent of the papers in this area. First, it has been shown that excitatory connections are like-to-like (with respect to orientation and/or direction preference), in papers such as Ko et al., Nature, 2011; Cossell et al., Nature, 2015; Lee et al., Nature, 2016; Wertz et al., Science, 2015; Rossi et al., Nature, 2020. Second, there are indications that inhibitory connections are also structured non-randomly with respect to neuron tuning properties (Znamensky et al., biorXiv, 2018). And Rossi et al., Nature, 2020, showed that inhibitory inputs to L2/3 E neurons form a pool that's shifted retinotopically relative to the pool of excitatory inputs. None of these features of connectivity are taken into account by the presented model. It is possible that a model with these known biological features would lead to different conclusions.

In fact, the results presented here contradict the results from the same group of authors published in Park et al., 2019, as they point out in Discussion:

"in vivo micro-ablation via laser of the apical tree in L2/3 mouse V1 pyramidal neurons did not abolish orientation selectivity, which remained essentially unchanged following a recovery period of ~ 1 day."

The authors explain this contradiction by potential homeostatic mechanisms "that might alter the dependence of somatic spiking on basal inputs". But given the points above, it seems equally likely that a more realistic model would have produced these effects. Since this observation is available from in vivo data, why not use it to constrain the model? That seems like the most natural thing to do, and I suspect modeling results with respect to the factors contributing to orientation selectivity would be then quite different.

3. Modeling inhibitory synapses as just noise, without any orientation tuning, is a limitation. Inhibitory neurons certainly show orientation tuning.

4. Overall, using constant-rate Poisson process for all synapses is suboptimal. Cortical neurons generally exhibit log-normal distributions of firing rates, meaning that a single rate for all inputs of a given type is far from realistic. Also, feedback inputs are typically delayed relative to feedforward inputs by about 10 ms; that would be important to take into account.

5. Orientation-tuned synapses should certainly exhibit different amounts of activity for different orientations. Here, synapses are always activated at 0.3 Hz, and their orientation preference is modeled by changing their weight depending on orientation. That is unrealistic. It is the presynaptic activity that changes depending on the orientation, and not the synaptic weight. (Weights can change due to plasticity, but the authors ignore plasticity here, and plasticity can work in either direction – depressing or facilitating – depending on the cell types.)

Some of the points above may seem like minor details, but they are not – consequences of such choices can be substantial. Such assumptions establish the correlation structure of synaptic inputs that is very different from what we know to happen in reality. Since the authors are investigating fine details of neuronal operation, including coincidence of basal and apical inputs, such effects are likely important for their study.

1. Both this manuscript and Park et al. compare the model voltage response to a somatic current injection with that from experiment (for a single cell), but no evidence is provided that electrophysiological properties of dendrites are captured well. In other words, we have no idea if currents and voltages in dendrites in response to synaptic inputs, local current injections, or somatic spikes or current injections, are captured well by the model. Capturing such properties well seems very important for the subject of the study, so, in the absence of a proof that it's working well, it is hard to trust any results from the model.

2. It's not clear why background-driven synapses all have the same activation frequency. Inhibitory synapses should be activated with higher rates, since many inhibitory neurons, especially PV fast-spiking neurons, fire at substantially higher rates than excitatory cells. The rates, 0.11 Hz for spontaneous and 0.3 Hz for orientation tuned inputs, seem too low. It is also not clear whether all synapses receive inputs from just two Poisson processes, or each synapse receives input from its own Poisson process. The former obviously creates unphysiological correlations between synapses, whereas the latter provides no correlations, which is also incorrect (it is well known that activity of neurons in the cortex is correlated – not perfectly, but substantially). Some middle ground informed by the data would be better than either of these extremes. While it may seem like an unimportant detail, in fact it is hugely important for the topic of the study, since correlations between incoming spikes will strongly influence the dendritic processing and coincidence detection.

3. Another big problem with using Poisson spike trains to simulate orientation selectivity is that it completely ignores the spatial structure of the stimulus. The model used only has stronger or weaker input depending on the orientation, but it's always Poisson random input. In reality, a neuron exposed, say, to a drifting grating, will be subject to alternating black and white stripes, which strongly modulate firing. The resulting firing pattern typically has periods when spikes follow each other in close succession and periods when there are barely any spikes at all. This temporal structure of synaptic activation, reflecting the spatial structure of the stimulus and retinotopic distribution of presynaptic neurons, is very likely to have strong effects on the postsynaptic neuron output and phenomena like dendritic integration.

I understand that my comments will be disappointing to the authors, and I am sorry for that. But I hope the comments might be helpful for their future work, and I am sure their approach will continue providing new interesting insights.

---

## [Author Response]

[Editors’ note: the authors resubmitted a revised version of the paper for consideration. What follows is the authors’ response to the first round of review.]

Reviewer #1 (Recommendations for the authors):This work aims to determine the contribution of apical and basal dendrites to orientation selectivity in a cortical neuron. This is a relevant question because basal and apical dendrites receive different qualitatively different classes of inputs: feedforward inputs favour basal dendrites whereas feedback inputs preferentially target apical dendrites. Understanding the rules by which these two dendritic fields interact at the level of single neurons is therefore useful for understanding and predicting how different information streams might be integrated by cortical neurons. Here, the authors take a compartmental modelling approach. Using a previously established model of a layer 2/3 pyramidal cell, they systematically vary some of the model parameters to explore and quantify how much apical and basal dendrites contribute to the generation of somatic spikes when synaptic input is tuned to orientation. The main conclusion is that both apical and basal contribute to spiking, with the apical contribution depending more on sodium conductances and the basal contribution depending mostly on AMPA and NMDA receptors. This conclusion is well supported by the data and the parameter exploration presented will be useful to the field. While the data come from a compartmental model, attempts to understand the biophysical basis of the phenomena described are limited and often not grounded in the current knowledge of cortical neuron biophysics. This aspect of the manuscript could be significantly expanded.Strengths:The main strength of this paper is the systematic variation of parameters such as synapse distribution and background activity, and the subsequent quantification of the effect of changing these parameters on somatic spiking. The disparity experiments are interesting, especially for seeing when orientation selectivity breaks down, and extend the modelling results presented in Park et al. 2019. The temporally precise manipulations of conductances and direct spike-by-spike comparison between control and manipulation conditions is a new and nice approach that should be useful for future studies. Another potential strength is the attempt to match the background activity observed in vivo, though this effort has significant weaknesses.

We thank the reviewer for highlighting the important contributions of our study.

Weaknesses:Most of the results in the paper describe the behavior of the model when a set of parameters are changed. While this is useful information, the major strength of a model is that it should be possible to understand exactly why model behaves the way it does, but the authors do not make a serious attempt at this. Why do apical dendrites rely more on sodium conductances to influence somatic spiking? Why do basals rely more on AMPA and NMDA conductances? What exactly is different in terms of the spatiotemporal patterns of inputs and voltage propagation across the neuron during "apical", "basal" and "cooperative" somatic spikes? The risk here is that the model behavior might only be valid for a restricted set of parameters (eg: density and distribution of conductances, most of which are not known experimentally), and thus the generality of the findings might be limited. Understanding the biophysics should allow for deriving general rules for the problem studied here and more robust conclusions.

We agree with the reviewer that this is an important point. To address it, we have performed several new simulations and analyses in order to provide explanations for our findings. Specifically, we performed a detailed statistical analysis whereby we link the morphological and electrophysiological characteristics of apical vs. basal dendrites to biophysical properties such as: (1) signal attenuation measured at the soma (both for subthreshold EPSPs and dendritic sodium spikes) and (2) the number of synapses (threshold) required to induce dendritic sodium spikes. This analysis focused at explaining why apical dendrites have a larger contribution to somatic output compared to basal dendrites, when considering sodium spikes (Figures 3 and 4). The results (see Section 3.5), shown in two new figures (Figure 5 and Supplementary Figure 8) revealed that signal attenuation is greater in apical than basal dendrites with similar anatomical characteristics. This is expected due to their highly branching structure and overall larger distance from the soma. Thus, signal attenuation cannot explain the higher contribution of apical dendrites to somatic output. However, the threshold for inducing sodium spikes in apical vs. basal dendrites (of similar anatomical characteristics) is significantly lower, making it easier to induce sodium spikes in the apical compared to the basal tree. This is due to the current sink effect of the soma which makes proximal dendrites – that is, mostly basal dendrites – harder to excite. This finding explains the higher contribution of the apical tree in driving somatic output.

In addition to the abovementioned results, we also performed a new sensitivity analysis whereby we modify the sodium channel conductance within each dendritic tree and show that our results remain unchanged for biologically relevant values (Supplementary Figure 7). This mitigates the risk that our findings only hold for a very strict parameter range.

Taken together, our findings support the conclusion shown in the Abstract Figure, whereby apical dendrites are proposed to contribute to somatic output through sodium spiking because they have a greater propensity to do so compared to basal dendrites, which suffer from a current sink effect due to their proximity to the soma.

Figure 6 is a step in this direction, but it mostly shows that the number of synapses to reach threshold increases with cable diameter and decreases with distance from soma, which is well known (ie: where the passive and active properties of the cable are uniform, as they are here, the result is explained by the differences in input impedance).

As mentioned in our response to the first comment, we extended this line of investigation by adding two new Figures (Figure 5 and Supplementary Figure 8) and discussing additional comparisons (i.e., mentioning the p-values of explored parameters that failed to yield significant differences between apical and basal dendrites). Furthermore, while it is indeed known that, all else being equal, the threshold “increases with cable diameter and decreases with distance from soma”, what we are concerned with here is whether this change in threshold (and also EPSP and spike attenuation) is the same between apical and basal dendrites, with the results of section 3.5 showing that it is not.

The lack of granularity is well illustrated in the title of section 3.6.: "Morphological and Electrophysiological Properties Influence Dendritic Behavior" – indeed they do, this is very well known. But how exactly in this model? Overall, I would have expected a more thorough analysis of the relationships between voltage propagation, morphology, and synaptic activation pattern. For example, could the main results here be explained by fast sodium spikes being more heavily filtered by the basal dendrites (which tend to have a shorter spatial constant, eg : Nevian et al. 2007), which would also explain the higher dependency of the basals on NMDA conductances, which are slower and therefore less filtered? Or are there differences in the relationship between synaptic input patterns and the generation of local non-linearities between basals and apicals?

We thank the reviewer for this comment, which we addressed with new simulations and analyses detailed in our previous responses.

Section 3.5 now mentions our findings related to sodium spike attenuation (as measured at the soma) in apical vs basal dendrites: “There was no significant difference in spike attenuation when normalized by distance from the soma (data not shown; p-value 0.768547), dendritic path volume (data not shown; p-value 0.061389), or ELC (data not shown; p-value 0.396982)”. Moreover, even without normalization, the extent of signal filtering is similar (slightly larger for apical dendrites) for both fast dendritic spikes and slow EPSPs across apical and basal dendrites. This is seen in Supplementary figure 8, whereby signal attenuation as a function of distance from the soma and branch point order is depicted. These new results demonstrate that the greater role of apical sodium spikes in neuronal spiking is not due to higher filtering of signals along basal dendrites.

Finally, synaptic activation patterns are unlikely to affect apical vs. basal contributions because they do not have a specific spatiotemporal pattern. Dendritic segments are activated by a large number of synaptic inputs (each receiving a different, independently sampled input spike train), which are distributed randomly but uniformly within the activated dendrites. All experiments are also repeated multiple times with different (randomized) input spike trains, to minimize the influence of any one particular input spike train. The setup of stimulus-driven input synapses is explained in our Extended Methods section, subsection 8.1.

The attempt to use experimental calcium imaging data to estimate background activity is nice, but the data are not very convincing. In Figure 2C, most of the isolated dendritic events look like noise. I realise that the event-triggered average shown in 3D suggests that there is some real signal being analysed, but I suspect that it is heavily contaminated by noise, and that the authors are grossly over-estimating the frequency of decoupled dendritic events. The method details of the analysis procedure are also worrying: "These percentile values were calculated via trial-and-error, attempting to match the expected firing frequency of a neuron (soma) under spontaneous activity conditions". What is the rationale here? The goal should be to detect the real events in the data, not changing the detection parameters until the result matches a previous publication. Also, if in the end the goal was to obtain the same value as other studies, it is unclear why the experiment was needed in the first place.

We thank the reviewer for this comment. We recognize that the data may not appear very convincing, but this is because of technical limitations beyond our control (e.g., low sampling frequency). Importantly, this data has no influence on the major findings of our study. Instead, we use this approach to assess the realism of our model, given the lack of data that can be used to reliably validate it. Although the trace shown in Supplementary Figure 2C might “look like noise”, it is vital to remember that events are detected via an unbiased algorithm whereby “a calcium event is registered when the ΔF/F value exceeds the 93.5^th^ percentile (nearly 2 standard deviations from the mean of a normal distribution) at the same time as the derivative of the ΔF/F value exceeds the 67^th^ percentile (1 standard deviation from the mean of a normal distribution)” – meaning that these events are not a commonly occurring “noise” signal, but a statistical outlier from the outset. In an attempt to (at least partially) account for the possibility of noise that could contaminate our event detection pipeline, we performed the Event-Triggered Average (ETA) analysis shown in Supplementary Figure 2D. The scaling of the dendritic vs somatic ETAs is also different (the two curves are not normalized), which could again mislead the viewer into thinking of the events as noise. We would like to offer Supplementary Figure 4 as additional evidence – although the increase in dendritic signal is seemingly not as prominent as in the somatic trace, note that the y-axis scaling is, again, different. We would also offer Supplementary Figures 2E and 2F as evidence – if the decoupled dendritic events were the result of noise, then surely the noise-free traces from the model data would not show any such events – but instead we find that the two (in vivo and model) match very closely. Regardless, we have removed these results from the main part of the manuscript as they do not influence our main conclusions and relegated them to a Supplementary Figure (Supplementary Figure 2), to de-emphasize their importance.

1. The authors propose that in Figure 1 they are validating their modelling approach, but it is unclear what exactly is being validated here. The figure shows that dendrites with sodium channels and AMPA/NMDA synapses can generate non-linear input-output curves, which is very much a guaranteed finding – in a model with reasonable passive parameters and morphology (which is the case here since model that has been previously validated against experimental data), I don't see how the result could be any different. Please clarify.

Unfortunately, it seems we may have failed to convey our intent in this section. We are not attempting to validate the model in this section (since it was already validated to the extent possible in Park et al., 2019), but rather to quantify the non-linear behavior of the dendrites, and compare the behavior of apical vs basal dendrites. Clearly the dendrites are non-linear, but the extent to which that is the case, as well as whether apical dendrites are more/less non-linear than basal dendrites is not clear or known a priori, and thus a valid target for exploration, and particularly important given our main research question. The distribution of Non-linearity Relative to Linear Extrapolation values (NRLE; Figure 1D) illustrates this most clearly – although all dendrites exhibit non-linear behavior, they do not feature identical NRLE values. We have also renamed this session to better convey our ultimate goal. It is now entitled “3.1. The degree of non-linearity varies across basal and apical model dendrites”.

2. Line 587: figure call is wrong, should be 4A (same for the subsequent call)

Thank you for spotting this. We have performed further proofreading of the manuscript, and have amended this, and many other, inaccuracies and typos.

3. In Figure 6 it is not clear why the results in panel C are qualitatively different that in A. Rm and Ra are presumably uniform, so the electrotonic constant is a transform of the dendritic diameter. Please explain.

Thank you for pointing this out. We accidentally omitted part of the calculation of the dendritic electrotonic length constant (section 8.5.8). Although the calculation of λ_i_ is correctly shown, we have failed to mention that the final value is calculated by dividing the length of each individual dendritic branch (L) by its λ_i_ value, which explains why there are substantial differences between these panels. We have amended this error in the new ‘Extended Methods’ section (section 8). That said, we have decided to remove these figures altogether, and present this data in a different way in Figure 5 (section 3.5).

Reviewer #2 (Recommendations for the authors):The manuscript by Petousakis et al. describes a modeling study of a pyramidal neuron in the Layer 2/3 (L2/3) of mouse primary visual cortex V1. The authors use a biophysically detailed, spatially extended model of the neuron, with voltage-gated conductances in the soma, basal dendrites, and apical dendrites to investigate how inputs to basal and apical dendrites sculpt the activity of the neuron and the corresponding major computation that L2/3 excitatory cells presumably carry out – that of orientation detection. The authors report a number of computational experiments and conclude that inputs to basal and apical dendrites synergistically sculpt the orientation selectivity of the neuron.While the premise is interesting, the study, and especially the model architecture, relied on a number of assumptions that limit the interpretability of the outcomes.1. The whole study is based on a single model of one neuron. This is a model from Park et al., Nature Communications, 2019, which in turn is based upon other previously published models (e.g., the supplementary tables 2-4 in this manuscript are copied from the Park et al. paper). Park et al. already showed how one can get orientation selectivity with this neuron model and how it can be manipulated by ablating apical and basal dendrites.

The use of a single morphology is indeed a limitation of our study, as recognized in the Discussion. However, the additional analyses that we performed in Section 3.5 (new Figure 5, Supplementary Figure 8) provide several new insights with respect to the morphology dependent reasons for which the apical/basal trees contribute differentially to somatic tuning. Therefore, one could use this morphological analysis to predict whether a different L2/3 V1 pyramidal neuron morphology is likely to behave like ours. In addition, while repeating our analysis using other morphologies is beyond the scope of this study (validating model function and performing the extensive parameter exploration reported here is extremely time-consuming), our code and all analysis scripts are made publicly available thus enabling interested scientists to apply our model/analysis to other morphologies. Finally, the sensitivity analysis that we added in Section 3.4.1 (Supplementary Figure 7) further establishes the robustness of our findings within biologically relevant ranges of ionic conductances. Overall, these new data and analyses demonstrate that our findings are novel and likely to generalize to other L2/3 V1 pyramidal neurons.

2. There is a lot of data on function-dependent connectivity for L2/3 excitatory neurons that sculpts orientation selectivity, but the authors do not mention even the most prominent of the papers in this area. First, it has been shown that excitatory connections are like-to-like (with respect to orientation and/or direction preference), in papers such as Ko et al., Nature, 2011; Cossell et al., Nature, 2015; Lee et al., Nature, 2016; Wertz et al., Science, 2015; Rossi et al., Nature, 2020. Second, there are indications that inhibitory connections are also structured non-randomly with respect to neuron tuning properties (Znamensky et al., biorXiv, 2018). And Rossi et al., Nature, 2020, showed that inhibitory inputs to L2/3 E neurons form a pool that's shifted retinotopically relative to the pool of excitatory inputs. None of these features of connectivity are taken into account by the presented model. It is possible that a model with these known biological features would lead to different conclusions.

We apologize for not clearly explaining the setup of our model. To clarify, the model does feature “like-to-like” connectivity (from the perspective of a single neuron), as stimulus-driven inputs are sampled from a non-uniform (Gaussian; see section 8.1 for details) distribution that features more synapses of an orientation preference closer to the orientation preference of the model neuron itself, effectively emulating “like-to-like” connectivity. The reviewer is correct, however, that our single-neuron model does not account for tuned or retinotopically shifted inhibitory inputs. While we recognize the important role of inhibition, these specific inputs cannot produce orientation tuning in L2/3 V1 neurons, and their increased activity can mask the effect of excitatory synapses and voltage-gated channels. Moreover, a retinotopic shift in inhibitory inputs would not affect our results, since stimulation is implicitly assumed to only take place at the center of the receptive field of the model neuron, a fact which would render any retinotopically offset neurons quiescent. Additionally, although inhibitory inputs could be biased towards one dendritic tree over the other, the net effect of such an imbalance would be a change in the net amount of excitation received by each tree – a scenario which we already explore in part (Figure 2 and Figure 3), finding tuning and somatic spike generation to be generally unaffected.

Thus, we decided to include a simple form of inhibition, whereby inhibitory synapses are background-driven, rather than stimulus-driven as was the case in Park et al., 2019. Although further investigation of the role of inhibition in orientation tuning is certainly very interesting, it lies beyond the scope of this work. We also opted for a relatively simple stimulation protocol, instead of the alternating stimulation of the “drifting gradient” protocol that is more widely used, for complexity reasons. Given that we performed a very extensive parameter exploration using a detailed compartmental model, the abovementioned simplifications were necessary in order to reduce the complexity (and thus computational requirements) of the model.

In fact, the results presented here contradict the results from the same group of authors published in Park et al., 2019, as they point out in Discussion:"in vivo micro-ablation via laser of the apical tree in L2/3 mouse V1 pyramidal neurons did not abolish orientation selectivity, which remained essentially unchanged following a recovery period of ~ 1 day."The authors explain this contradiction by potential homeostatic mechanisms "that might alter the dependence of somatic spiking on basal inputs". But given the points above, it seems equally likely that a more realistic model would have produced these effects. Since this observation is available from in vivo data, why not use it to constrain the model? That seems like the most natural thing to do, and I suspect modeling results with respect to the factors contributing to orientation selectivity would be then quite different.

We apologize for not communicating our model findings more clearly, as this comment is incorrect. Given that our model is near-identical to the Park et al., 2019 model (identical barring the addition of a 10 ms delay to stimulus-driven apical synaptic inputs, to emulate delayed feedback activity), it would be impossible for the model to contradict these prior results. To demonstrate this, we have reproduced the results of the ablation experiment from Park et al., 2019 using our current model (Supplementary Figure 9) and have added new text in the Discussion section of the manuscript to explain that this is the case. There is no contradiction between the Park et al. findings and this work.

3. Modeling inhibitory synapses as just noise, without any orientation tuning, is a limitation. Inhibitory neurons certainly show orientation tuning.

Indeed, this is a limitation of our model – but one we believe has a minor impact in our simulations as inhibition is uniformly distributed and untuned throughout the basal and apical trees. The same form of inhibition was also used in the Park et al., 2019 study. We have expanded upon our reasoning for this choice in our response to point #2, as well as in the Discussion section of the manuscript.

4. Overall, using constant-rate Poisson process for all synapses is suboptimal. Cortical neurons generally exhibit log-normal distributions of firing rates, meaning that a single rate for all inputs of a given type is far from realistic. Also, feedback inputs are typically delayed relative to feedforward inputs by about 10 ms; that would be important to take into account.

We thank the reviewer for this important point and apologize for the lack of clarity in presenting our model. In fact, our model features *variable*-rate Poisson processes for all stimulus driven synaptic inputs, and these Poisson processes are different (different sample) for each synapse, even if the rates happen to be the same. Unfortunately, our description in the methods was severely lacking here, very easily misleading the reader into believing that what was being varied was the *weight* of the synapses, which is *not* the case. The process being followed is that values of the weight vector are being used to re-scale a base activation rate (0.3 Hz) for the Poisson processes of stimulus-driven synapses, leading to variable-rate Poisson processes for synapses of different orientation preferences (verifiable by inspecting the model file “stimulus_uniform_basalvar.hoc”). This only applies for stimulus-driven excitatory synapses, and is not the case for background-driven inhibitory and excitatory inputs. More details can be found in the Extended Methods, section 8.1. As for the incorporation of delayed feedback, we concurred that it would be a very important addition, which is why we incorporated it into all apical stimulus driven synapses in the model. We redid all experiments that would be affected by this change, but there was no significant change in our overall results (e.g. see sections 3.2, 3.3 and 3.4).

5. Orientation-tuned synapses should certainly exhibit different amounts of activity for different orientations. Here, synapses are always activated at 0.3 Hz, and their orientation preference is modeled by changing their weight depending on orientation. That is unrealistic. It is the presynaptic activity that changes depending on the orientation, and not the synaptic weight. (Weights can change due to plasticity, but the authors ignore plasticity here, and plasticity can work in either direction – depressing or facilitating – depending on the cell types.)

Again, we apologize for the lack of clarity. As mentioned in our response to the previous point, this was a misunderstanding borne out of a lacking explanation from our part – synapses are *not* always activated at 0.3 Hz, and the orientation preference is modeled by a frequency change, not a weight change. The “weight” was merely a vector used to rescale the activation frequency. Plasticity was indeed not explored in this work – despite being an important process, it was outside the scope of our experimentation.

Some of the points above may seem like minor details, but they are not – consequences of such choices can be substantial. Such assumptions establish the correlation structure of synaptic inputs that is very different from what we know to happen in reality. Since the authors are investigating fine details of neuronal operation, including coincidence of basal and apical inputs, such effects are likely important for their study.1. Both this manuscript and Park et al. compare the model voltage response to a somatic current injection with that from experiment (for a single cell), but no evidence is provided that electrophysiological properties of dendrites are captured well. In other words, we have no idea if currents and voltages in dendrites in response to synaptic inputs, local current injections, or somatic spikes or current injections, are captured well by the model. Capturing such properties well seems very important for the subject of the study, so, in the absence of a proof that it's working well, it is hard to trust any results from the model.

A fair point raised by the reviewers was the lack of convincing evidence concerning the dendritic biophysics of the model. Unfortunately, there is little to no data on dendrites of L2/3 V1 neurons, making a validation process more precise than the one followed in the setup of the original model (Park et al., 2019) unfeasible. This is true for all L2/3 V1 models, and yet several of them are published and extensively used by the community, highlighting the importance of building even partially validated models for moving the field forward (e.g. Goetz, Roth and Haüsser, 2021). In order to alleviate the concern that our results may be due to the selected value for VGSC conductance, we performed experiments where apical VGSC conductance was reduced, and the impact of this reduction evaluated (Supplementary Figure 7). Results show that our findings are robust to changes in conductance up to a point, beyond which the firing rate of the neuron is too low to adequately represent a L2/3 V1 neuron (see section 3.4). In addition, we have attempted to capture the spatiotemporal evolution of voltage propagating through the neuron by including an extra set of plots that show how apically- and basally-driven spikes are initiated and propagate through the cell (Supplementary Figure 6). Finally, we have greatly expanded the last section of our results (3.5), allowing us to provide more weight to our findings concerning the different “mechanisms of action” of apical and basal dendrites. We hope that this additional evidence will add sufficient weight to our modeling and findings.

2. It's not clear why background-driven synapses all have the same activation frequency. Inhibitory synapses should be activated with higher rates, since many inhibitory neurons, especially PV fast-spiking neurons, fire at substantially higher rates than excitatory cells. The rates, 0.11 Hz for spontaneous and 0.3 Hz for orientation tuned inputs, seem too low. It is also not clear whether all synapses receive inputs from just two Poisson processes, or each synapse receives input from its own Poisson process. The former obviously creates unphysiological correlations between synapses, whereas the latter provides no correlations, which is also incorrect (it is well known that activity of neurons in the cortex is correlated – not perfectly, but substantially). Some middle ground informed by the data would be better than either of these extremes. While it may seem like an unimportant detail, in fact it is hugely important for the topic of the study, since correlations between incoming spikes will strongly influence the dendritic processing and coincidence detection.

As mentioned previously, this was a misunderstanding borne out of a lacking explanation from our part – synapses are not always activated at 0.3 Hz, and the orientation preference is modeled by a frequency change, not a weight change. Our work does not explicitly model different interneuron subtypes, and purposely simplifies modeling of inhibitory inputs in order to alleviate computational load (see also our response to point #4 above). Each synapse samples an input spike train from a Poisson distribution independently. Input correlations are not enforced, but emerge as a side-effect of the large number of spike trains sampled from similar distributions.

3. Another big problem with using Poisson spike trains to simulate orientation selectivity is that it completely ignores the spatial structure of the stimulus. The model used only has stronger or weaker input depending on the orientation, but it's always Poisson random input. In reality, a neuron exposed, say, to a drifting grating, will be subject to alternating black and white stripes, which strongly modulate firing. The resulting firing pattern typically has periods when spikes follow each other in close succession and periods when there are barely any spikes at all. This temporal structure of synaptic activation, reflecting the spatial structure of the stimulus and retinotopic distribution of presynaptic neurons, is very likely to have strong effects on the postsynaptic neuron output and phenomena like dendritic integration.

This model does indeed ignore the spatial structure of the stimulus, and does not model retinotopic effects (see also our response to point #2 above). It is true that in a “drifting grating” paradigm, the firing rate of the neuron during the shift from “dark” to “light” (or vice-versa) can be much higher than the observed peak mean firing rate of our model neuron. Although this would clearly influence the moment-to-moment output of the neuron, this would only be relevant in our evaluation of its orientation tuning capabilities (since no other neurons are modeled). As for the influence of this higher instantaneous firing rate on dendritic integration, our model already features dendrites producing sodium spikes at rates well in excess of the experimentally observed firing rate of L2/3 V1 pyramidal neurons, as well as backpropagation of somatic action potentials. As such, we do not believe that an increase in the instantaneous (but not the average) firing rate of the neuron would significantly change our results.